# Stochastic Forward-Forward Learning through Representational Dimensionality Compression

**Zhichao Zhu**[1, 2]
zhichao_zhu@fudan.edu.cn

**Yang Qi**[1,2,3]
yang_qi@fudan.edu.cn

**Hengyuan Ma**[1, 2]
hangyuanma21@m.fudan.edu.cn

**Wenlian Lu**[1,2,4]
wenlian@fudan.edu.cn

**Jianfeng Feng**[1,2,3,*]
jffeng@fudan.edu.cn

## Abstract

The Forward-Forward (FF) learning algorithm provides a bottom-up alternative to backpropagation (BP) for training neural networks, relying on a layer-wise "goodness" function with well-designed negative samples for contrastive learning. Existing goodness functions are typically defined as the sum of squared postsynaptic activations, neglecting correlated variability between neurons. In this work, we propose a novel goodness function termed dimensionality compression that uses the effective dimensionality (ED) of fluctuating neural responses to incorporate second-order statistical structure. Our objective minimizes ED for noisy copies of individual inputs while maximizing it across the sample distribution, promoting structured representations without the need to prepare negative samples. We demonstrate that this formulation achieves competitive performance compared to other non-BP methods. Moreover, we show that noise plays a constructive role that can enhance generalization and improve inference when predictions are derived from the mean of squared output, which is equivalent to making predictions based on an energy term. Our findings contribute to the development of more biologically plausible learning algorithms and suggest a natural fit for neuromorphic computing, where stochasticity is a computational resource rather than a nuisance. The code is available at `https://github.com/ZhichaoZhu/StochasticForwardForward`.

## 1 Introduction

Despite being central to the success of traditional deep learning, backpropagation (BP) poses challenges for on-chip learning in neuromorphic systems, as it requires global error signals and symmetric weight transport, both of which are widely regarded biologically implausible and difficult to implement efficiently on neuromorphic hardware [1–5]. Therefore, the forward-forward (FF) learning algorithm proposed by Hinton [6] provides an elegant bottom-up alternative that each layer learns independently by maximizing a "goodness" measure of its activations, eliminating the need for error backward propagation.

Although FF learning is conceptually simple, its success relies on generating high-quality negative samples, which is highly task-specific and presents a significant practical challenge. Moreover, the goodness function in the original FF learning is the sum of squared postsynaptic activations and it does not account for the role of noise, a ubiquitous feature in both biological neural systems [7, 8] and

---

[1]Institute of Science and Technology for Brain-Inspired Intelligence, Fudan University, Shanghai 200433, China.[2]Key Laboratory of Computational Neuroscience and Brain-Inspired Intelligence (Fudan University), Ministry of Education, China. [3]MOE Frontiers Center for Brain Science, Fudan University, Shanghai 200433, China. [4]Ji Hua Laboratory, Foshan 528200, China. *Corresponding author.

39th Conference on Neural Information Processing Systems (NeurIPS 2025).

neuromorphic computing hardware [9, 10]. Substantial evidence suggests that neuronal correlated variability could carry rich information [11–13] and can be considered a computational resource [14–18]. Leveraging noise and incorporating postsynaptic neuron correlation into FF learning could potentially lead to more biologically plausible and hardware-adapted alternatives.

In this work, we extend the FF framework by introducing a novel goodness function termed dimensionality compression that is derived from the effective dimensionality (ED) [19] of neuronal responses. Essentially, ED depicts the second-order statistical structure of neural responses, which can be used to describe representational selectivity. Within-class responses should have low ED, indicating that samples are tied to a specific variable, while across-class responses should have high ED to avoid representation collapse. Instead of generating negative samples, we introduce noise to create isologues of the samples, with the objective of minimizing ED for each sample while maximizing it for all inputs. This promotes locally structured representations that are both compact and discriminative, bypassing the need for negative samples and fitting smoothly into the FF training pipeline. The numerical experiments on standard datasets demonstrate that the proposed method can achieve performance comparable to that of other non-BP methods. Furthermore, this approach inherently recommends using the mean of squared outputs rather than just averaging them for prediction, which means that variability in the outputs can also carry the information interested and can be interpreted as an extension of energy-based learning (EBL) [20–22]. Collectively, our findings provide both a theoretical link between dimensionality and representation learning and a practical direction toward biologically plausible, noise-driven computation.

## 2 Background and related works

We begin by reformulating the learning problem of a network in classification tasks. Given an input-target pair $(X, t)$ from $T$ targets, the input is processed through $L$ blocks with $X^{(l)} = f(X^{(l-1)}, \theta^{(l)})$ and $X^{(0)} = X$. A linear classifier $W$ then produces class scores $Y = X^{(L)} W^T$ and denotes $Y_i$ the score for the target $i$. The question posed is how to modify the model parameters in the absence of BP, while still guaranteeing that the output $Y_t$ associated with the true target $t$ achieves maximal discrimination.

Since learning performance is ultimately assessed through a linear classifier, it is intuitive to assume that if each block produces more discriminative representations, stacking such blocks can improve overall classification. For example, direct feedback alignment (DFA) [23] demonstrates that deep networks can learn effectively without requiring symmetric feedback connections. Instead, DFA delivers global error signals to each layer through fixed random feedback pathways, showing that precise gradient transmission is not essential for credit assignment.

While DFA relaxes the need for symmetric feedback by transmitting error signals through random fixed pathways, EBL [20–22] eliminates the requirement to propagate explicit errors altogether. Instead, EBL frames learning as minimizing an energy function that depends on local neuronal interactions. During training, the network first clamps the input and fixes the output to the target, allowing neural activities to evolve toward an equilibrium state that minimizes this energy. Weight updates are then computed locally based on the pre- and post-synaptic activities. However, because the energy landscape of one layer depends on others, finding a global equilibrium in deep architectures can be computationally demanding and slow to converge.

One can simplify the interaction between blocks by training the blocks independently from the bottom up. Greedy InfoMax (GIM) [24] optimizes individual blocks by encouraging each block to preserve information about its input, which can be interpreted as reinforcing each block to learn slow features [25] from its inputs, allowing more scalable unsupervised representation learning without global backpropagation. Hinton's FF algorithm[6] offers a simpler energy-based approach, using a scalar goodness function to distinguish 'positive' and 'negative' samples in each layer. Initially, this function, similar to the energy function in EBLs, is the square of postsynaptic neuronal activities. However, it is necessary to use high-quality negative samples for contrast learning, as optimizing the network with this function alone will fail because it encourages maximum neuron activation. Hinton also noted that binary objectives consisting of one "positive" and one "negative" sample inject limited information per update, which will slow the rate at which meaningful structure can be encoded in the weights.

Recent advancements in FF-inspired algorithms have removed the requirement for negative samples by employing a supervised learning approach. The Cascaded Forward (CaFo) model [26] enhances the FF algorithm through the integration of convolutional layer blocks, allowing these blocks to independently generate label distributions without requiring negative samples. Papachristodoulou et al. [27] further subdivided the channels within the convolutional layers into $T$ groups, optimizing the mean activation of each group to be more pronounced for certain classes, thus improving linear separability. While these modifications can be effective, implementing such strong label-based supervision in a biologically brain can be challenging, as it necessitates knowing a label for each input pattern.

Hebbian learning is acknowledged as a biologically realistic mechanism for synaptic plasticity, enabling weight adjustments in an unsupervised manner. However, the principal obstacle lies in its limited efficacy, particularly when deployed in large networks and more complex datasets. Although Journé et al. [28] and Nimmo and Mondragon [29] showed that combining unsupervised Hebbian learning with a well-designed winner-take-all (WTA) mechanism yields satisfactory results, their success involves complex softmax activation functions combining with several optimization tricks. These complications hinder a deeper understanding of the core question: What should a neural block learn when limited to local information and how to define the goodness of neuronal response for goal achievement?

In this study, we aim to answer these questions by leveraging the continuous, second-order structure in neural responses, providing a more efficient and biologically grounded alternative to contrastive objectives.

## 3 Effective dimensionality as a goodness function

Historically, the effective dimensionality (ED) is introduced to describe the equivalent number of orthogonal dimensions that would produce the same overall pattern of covariation of a set of correlated variables [19]. In practice, ED is typically based on the covariance matrix of the data [30, 31]. However, in neural or network representations where the mean activity itself encodes task or stimulus information, it is more appropriate to consider the uncentered second moment as it reflects both the signal (mean configuration) and interaction structure (correlation). Therefore, in this work, we define

$$\text{ED}(X^{(l)}) = \frac{\text{tr}(\mathbb{E}[X^{(l)T}X^{(l)}])^2}{\|\mathbb{E}[X^{(l)T}X^{(l)}]\|_F^2} = \frac{(\sum_{i=1}^d \lambda_i)^2}{\sum_{i=1}^d \lambda_i^2}, \tag{1}$$

where $\lambda_i$ are the eigenvalues of the uncentered covariance matrix of neural activity $X^{(l)}$. If $X^{(l)}$ has zero mean, this definition reduces to the standard ED based on the covariance matrix that gives a good indication of the number of principal components needed to capture most of the variance of $X^{(l)}$ (Fig. 1**a**). When then mean is nonzero, ED based on $\mathbb{E}[X^{(l)T}X^{(l)}]$ not only measures the diversity of local variations but also incorporates the global structure imposed by the mean configuration. By varying the mean and covariance of a two-dimensional distribution, we illustrate that ED is maximized for distributions whose dimensions are uncorrelated with identical mean and variance but will reduce otherwise (Fig. 1**b**).

In particular, $\mathbb{E}[X^{(l)T}X^{(l)}]$ is also a valid representation of the energy landscape of the $l$-th block. In conventional EBL, the energy term is typically defined as $\text{tr}(\mathbb{E}[X^{(l)T}X^{(l)}])$ that quantifies the total expected energy and the overall magnitude of population activity while neglecting the correlation structure between neurons. In contrast, ED incorporates the full spectral structure of $\mathbb{E}[X^{(l)T}X^{(l)}]$, thus providing a correlation-aware measure of the energy landscape. In this sense, ED-based learning can be regarded as an extension of EBL that preserves the connection to energy while emphasizing the structural geometry of the representation.

To illustrate this concept, we start by considering a population of neurons where each neuron is selectively responsive to a particular type of stimuli (Fig. 1**c**) and where the response of the neuron conditioned on a specific input is a function of some stimulus features $s$, commonly referred to as a tuning curve in neuroscience [12]. Consider a representational space spanned by two neurons in Fig. 1**c** that are selectively activated for a specific feature $s$ informative about the input class. Then, neural responses to inputs within the same class are expected to be more aligned along a specific direction (Fig. 1**d**). In such an ideal case, the ED for within-class responses (blue and orange

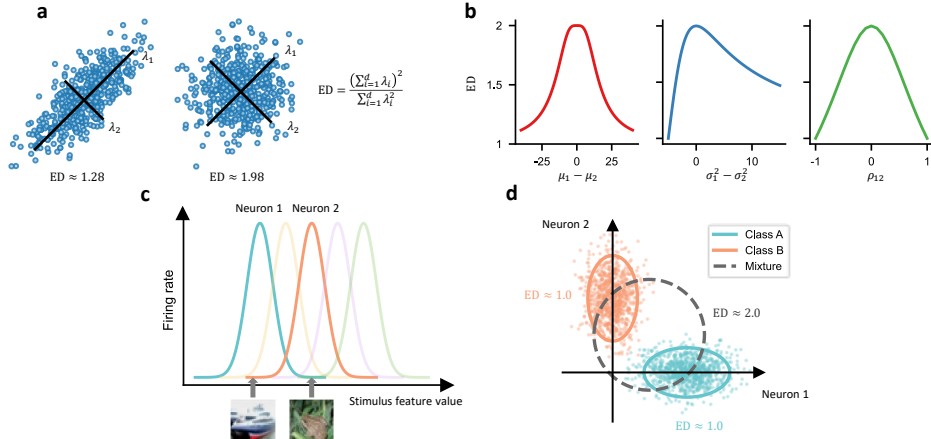

Figure 1: **Assessing the informativeness of neuronal responses through effective dimensionality (ED). a.** Illustration of what ED quantifies for a zero-mean Gaussian distribution. The eigenvalues $\lambda_i$ computed from the uncentered second moment. ED approaches 1 when variance is concentrated along a single principal direction (left) and increases toward 2 as variance becomes isotropic (right). **b.** Influence of mean and covariance on ED. In the left and middle panels $\mu_2 = 0, \sigma_1^2 = \sigma_2^2 = 1$ and $\mu_2 = \mu_1 = 0, \sigma_2^2 = 5$ are fixed while varying $\mu_1$ and $\sigma_1^2$ respectively. In the right panel, $\mu = 0, \sigma^2 = 1$ are fixed while varying the correlation coefficient $\rho_{12}$. **c.** Example tuning curves showing neurons selectively responsive to some category-informative features, forming a population code that encodes categorical information. **d.** ED as a measure of class separability in a two-dimensional response space. Points represent noisy samples from two classes (blue and orange). Within-class responses form clusters with low ED, whereas their mixture (whose uncentered covariance is represented by the dashed gray ellipse) exhibits higher ED, reflecting representational diversity.

dots) is expected to be close to one, indicating that inputs can be well explained by fewer variables compared to the dimensionality of the representational space. In contrast, the ED for the mixed responses across classes (whose uncentered covariance is represented by the dashed gray circle) is expected to approach two. Therefore, inputs belonging to different classes are explained by unique variables orthogonal to each other, leading to a linearly separable representation. The learning goal of ED-based learning is thus straightforward: we should minimize ED for responses within each class to promote consistency and robustness, while maximizing ED across all inputs to ensure diversity and discriminability.

The problem is, the label information is infeasible in the unsupervised setting. To compute ED in the absence of true class labels, we introduce noise into the computing process, which is an essential characteristic of both biological systems and neuromorphic hardware [7, 14, 17, 18, 10].

A crucial point is that introducing moderate noise does not hinder the identification of its class (Fig. A1). In other words, the essential features that form the concept of its class remain intact. This strategy enables unsupervised learning of robust and discriminative features, but bypasses the need for explicit class labels or negative sampling. Formally, let $X' \in \mathbb{R}^{B \times F}$ be a batch of data samples and use dropout to create its noisy copies, $X$, by randomly setting the elements in $X'$ to zero with probability $p$. For the $l$-th block, we denote its output (i.e. the corresponding neural responses) as $X^{(l)}$ and $X_i^{(l)}$ are the neural responses to a particular input sample in the batch. The objective function for each block is then consisting of a consistency term and a diversity term, defined as:

$$\text{ED}_c = \frac{1}{B} \sum_{i=1}^{B} \text{ED}\left(X_i^{(l)}\right), \tag{2}$$

and

$$\text{ED}_d = \text{ED}\left(\mathbb{E}[X^{(l)}]\right) \tag{3}$$

where $B$ is the batch size, $\mathbb{E}[X^{(l)}]$ denotes the averaging of noisy copies of the input batch. $\text{ED}_c$ describes the mean ED of neural responses to noisy realizations of individual input samples and

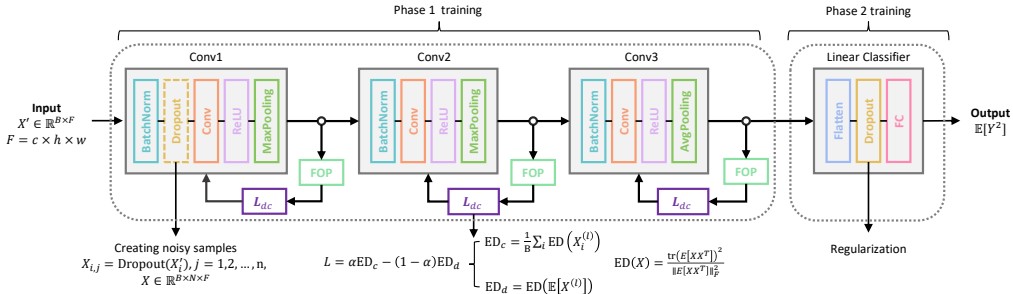

Figure 2: **Network architecture and training pipeline.** The first dropout layer generates $N$ noisy variants per input and remains active during inference, while the dropout in the linear classifier is used only for regularization during training. Batch normalization (BN) layers stabilize inputs and contain no trainable parameters. Each convolutional block includes a Fixed Orthonormal Projection (FOP) module that projects its output onto a subspace with a pregenerated random orthonormal basis before computing the dimensionality compression loss $L$. Training proceeds in two phases: (1) Each convolutional block is trained layer-wise for 3 epochs using the proposed loss function $L$. (2) The convolutional blocks are then frozen, and a linear classifier is trained for 60 epochs using cross-entropy loss, where prediction score for each sample is computed as the mean of squared classifier outputs over the noisy variants. The overall architecture and training pipeline are consistent across all experiments, except for the classifier's input dimensionality, which varies by dataset.

captures the noise variability, whereas $\text{ED}_d$ describes the ED of the response distribution in different input samples and captures the variability of the data.

The dimensionality compression loss is then formulated by merging the two components, with a trade-off parameter $\alpha$ that is assigned a default value of 0.5.

$$L = \alpha\text{ED}_c - (1 - \alpha)\text{ED}_d. \qquad (4)$$

Conceptually, minimizing $\text{ED}_c$ implicitly suggests a WTA dynamic [28, 29] so that the responses are more distinct in some directions compared to others, leading to a more compact representation. In contrast, maximizing $\text{ED}_d$ prevents the collapse of representation as it reinforces the overall responses that can be explained by as many directions as possible. In this way, the network is encouraged to learn class-discriminative features, leading to a more linearly separable representation.

Notably, this learning objective also motivates a modification of the inference procedure. Let $Y_i$ be the output of the linear classifier for a noisy input sample $X_i$. Rather than averaging the output, we propose using $\mathbb{E}[Y_i^2]$ as the classification score. This approach is equivalent to selecting the neuron that has the minimum energy, as inline with EBL.

## 4 Experimental design and results

**Architecture.** As shown in shown in Fig. 2, we use a network architecture similar to that used by Journé et al. [28] and Nimmo and Mondragon [29] except the activation function chosen to verify the effectiveness of the proposed goodness function. The network consists of three convolutional blocks followed by a fully connected layer for classification. The first block has 96 channels, and the number of channels increases by 4 for each subsequent layer. Each channel is regarded as a neuron due to weight sharing, and the corresponding feature map is treated as samples drawn from an unknown distribution conditioned on the input. See Appendix A1.1 for details.

**Datasets and data preprocessing.** We use the MNIST[32], CIFAR-10 and CIFAR-100 datasets [33] to verify the effectiveness of the proposed goodness function. For MNIST, we only use random crop for data augmentation. For CIFAR-10 and CIFAR-100, we first apply zero phase component whitening and then random crop and random horizontal flip for data augmentation.

**Training Pipeline.** We divide the training process into two phases. In the first phase, convolutional blocks are trained using the proposed goodness function layer-wise for 3 epochs. Specifically, for each epoch, we train the first block with the proposed loss $L$, fix it, and then train the next block.

Table 1: Comparison of validation accuracy (%) of the proposed method against recent non-BP and FF-inspired approaches, along with BP, that adhere to our architecture across various datasets. The performance of the proposed method are estimated over 5 runs.

| Method | Validation Accuracy (%) | | |
|---|---|---|---|
| | MNIST | CIFAR10 | CIFAR100 |
| BP | $99.33 \pm 0.04$ | $82.50 \pm 0.09$ | $61.28 \pm 0.25$ |
| Original FF [6] | 98.73 | 59 | - |
| CaFo FF [26] | 98.95 | 69.49 | 42.13 |
| CwC FF [27] | $99.42 \pm 0.08$ | $78.11 \pm 0.44$ | 51.32 |
| DFA [23] | $98.98 \pm 0.05$ | $73.10 \pm 0.50$ | $41.00 \pm 0.3$ |
| Soft Hebbian [28] | $99.35 \pm 0.03$ | $80.31 \pm 0.14$ | 56.00 |
| Hard Hebbian [29] | - | 76 | - |
| GIM* [24] | $99.29 \pm 0.03$ | $78.19 \pm 0.34$ | $50.09 \pm 0.45$ |
| EBL [21] | 99.56 | 89.6 | 65.8 |
| Proposed method | $99.31 \pm 0.07$ | $76.96. \pm 0.73$ | $53.29 \pm 1.02$ |

\* Reimplementation results.

This process is repeated for all blocks. Given that the number of neurons far exceeds both the number of samples and the number of classes, direct estimation of ED may become unreliable due to the sparsity of samples. To address this, we propose projecting the output of each block onto a lower-dimensional subspace using a randomly generated set of orthogonal basis vectors before computing the dimensionality compression loss. This projection operation is expected to reduce the estimation variance while preserving the structural properties of the representation. We gradually reduce the projection dimensionality until it aligns with the number of output classes.

In the second phase, convolutional blocks are frozen, and we train the linear classifier for 60 epochs with the standard cross-entropy loss, where the prediction score is based on the mean of squared outputs over the noisy variants. The best validation accuracy is reported as we care more about the upper bound of the performance. All experiments utilize a NVIDIA RTX 3090 GPU and an Intel Xeon(R) Gold 6226R CPU, as detailed in the Appendix A1.2.

## 4.1 Proposed method achieves comparable performance with other non-BP methods

As the development of non-BP methods is still in its infancy, researchers use different network architecture, training protocols, and datasets to evaluate the performance of their methods, making it difficult to make a fair comparison. Therefore, we use other non-BP methods' results reported in their papers for comparison. The results are summarized in Table 1.

We find that the proposed method achieves comparable performance with other non-BP methods on MNIST and CIFAR-10 datasets, and show its ability to learn useful features in CIFAR100. In the realm of FF methods, the work by Hinton [6] is seen as the baseline. CaFo FF [26] modifies the initial goodness function by independently training each convolutional block for classification. Our method consistently surpasses these two methods in all datasets, as well as the DFA proposed by Nøkland [23]. For CwC FF [27] which revises the convolutional framework by dividing the channels into $C$ groups for supervised learning, our method can match its performance without labels and specific requirements for network architectures. Given that our network architecture mirrors that of Hebbian learning combined with soft or hard WTA mechanisms [28, 29], our results are directly comparable to those works and attain similar performance levels. In addition, as previously noted, optimizing ED naturally introduces a competition mechanism, which can be considered as a general learning principle that these methods strive to achieve. The GIM [24] also achieves a performance similar to that of our method when reimplemented using this network architecture (see Appendix A1.2).

For EBL methods, Scellier et al. [21] conducts a comparative study of existing EBL approaches, utilizing a five-layer convolutional Hopfield network to demonstrate that equilibrium-backpropagation [20] achieves superior performance. Although our findings are inferior compared to theirs, the discrepancies may be attributed to the deeper network used and the ability of EBLs to leverage top-down

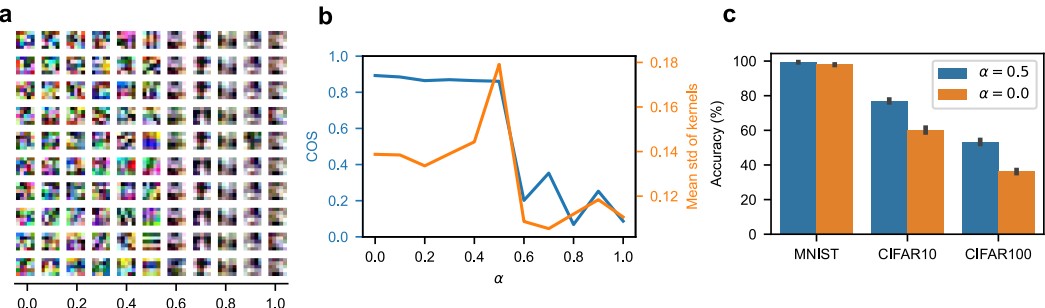

Figure 3: **Effect of the trade-off factor $\alpha$ on weight optimization. a.** Visualization of the first-layer convolutional kernels trained on CIFAR-10 under different values of $\alpha$. Each column shows the top 10 channels ranked by the standard deviation of their weights trained with different $\alpha$. **b.** Cosine orthogonality score (COS, blue line) and mean standard deviation of first-layer kernels (orange line) as functions of $\alpha$. A higher COS indicates greater diversity among channels' weights. **c.** Classification accuracy comparison for $\alpha = 0.0$ and $\alpha = 0.5$ (default). Error bars denote one standard deviation across 5 independent training runs.

information for optimization. Unfortunately, adding more than three convolutional blocks in our experiments leads to a performance drop, which we attribute to the difficulty in preventing the collapse of class-specific features after a stack of highly nonlinear transformation, a technical problem that needs to be solved in future works.

In contrast to BP under the same architecture and training protocol, the proposed method performs similarly on MNIST. However, the disparities widen with CIFAR-10 and CIFAR-100, a frequent issue for non-BP methods. Since the power of BP comes from its ability to optimize a network globally so that the final output ultimately meets the task's requirements, it leaves room for future work to explore how top-down and bottom-up learning can be combined in a more biological plausible manner, thus making the implementation on hardware more friendly.

Since one distinct feature of our method is the introduction of noise during training, we also conduct experiments by varying the noise strength (changing the dropout rate $p$) and sampling size to assess its impact on performance in the Appendix A1.3. These results demonstrate that both noise strength and sampling size significantly influence model performance. In general, moderate noise levels and appropriate sampling sizes yield optimal results.

In summary, the proposed method achieves a performance comparable to that of other non-BP methods and even outperforms some of them. Although we currently are unable to demonstrate the efficiency of our method in deeper networks or more complex datasets, the use of noise and the simplicity of the learning objective make it a promising direction to explore how noise can facilitate learning in a biologically plausible manner.

## 4.2 Compressing dimensionality leads to orthogonal weights

We next train the first convolutional blocks on CIFAR10 by varying $\alpha$ from 0 to 1 with a step increase of 0.1 to investigate the effect of the trade-off factor $\alpha$ on weight optimization. Each column in Fig. 3**a** displays the weights of the ten channels with the highest standard deviation in their weights among all channels trained under a given $\alpha$. The plots indicate a transition point at $\alpha = 0.5$. When $\alpha > 0.5$ is applied for training, the channel weights collapse into a similar pattern, suggesting the limited capacity of the layer to learn a varied combination of input features. In contrast, when $\alpha \leq 0.5$, diverse weight patterns emerge.

To quantify this observation, we compute the cosine orthogonality score (COS) of the kernels, defined as

$$\text{COS} = \frac{1}{K} \sum_{j < i} \left( 1 - \left| \frac{\langle w_i, w_j \rangle}{\|w_i\| \|w_j\|} \right| \right), \tag{5}$$

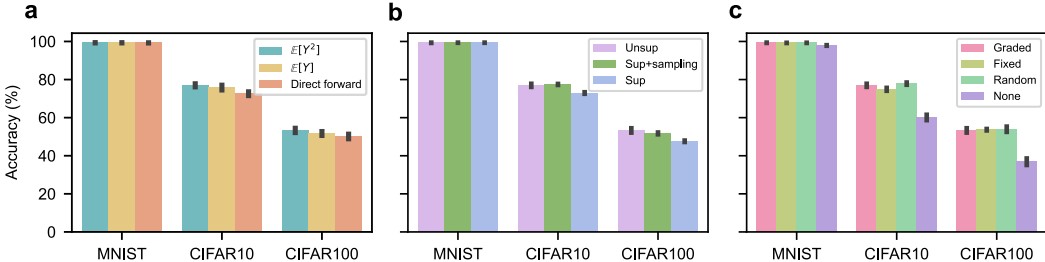

Figure 4: **Factors affecting task performance. a.** Classification accuracy under different inference strategies. $\mathbb{E}[Y^2]$: proposed method, using the mean squared outputs (energy) based on generated noisy samples as prediction score; $\mathbb{E}[Y]$: uses the mean of outputs as prediction score. *Direct forward*: standard inference without noise, using raw inputs. **b.** Accuracy under different training schemes. *Unsup*: proposed method, where $ED_c$ is computed at the instance level based on generated noisy samples. *Sup+sampling*: generated noisy samples are further grouped by class labels before computing $ED_c$. *Sup*: computes $ED_c$ directly on labeled data without the need to generate noisy samples. **c.** Accuracy under different projection strategies. *Graded*: block outputs are projected with gradually decreasing dimensions (30-20-10 for MNIST and CIFAR-10; 90-150-100 for CIFAR-100). *Fixed*: all blocks projected to a constant dimension equal to the number of classes. *Random*: projected to a randomly selected dimension per block. *None*: no projection.

where $w_i$ is the $i$-th channel and $K = \frac{c(c-1)}{2}$ is the number of pairs {i,j} with $c$ being the number of channels. As shown in Fig. 3**b**, the COS (blue line) is consistently close to one if $\alpha \leq 0.5$ , indicating that the kernels are fully orthogonal to each other. When $\alpha > 0.5$, the COS decreases significantly and indicates that the weights of different channels tend to be similar, leading to highly redundant feature extraction. This is consistent with the observation in Fig. 3**a** that all the weights of the channels are similar to each other when $\alpha > 0.5$.

This transition point is also observed when using the mean standard deviation of the channels (Fig. 3**b**, orange line). When $\alpha > 0.5$, the average standard deviation decreases substantially, suggesting that the numerical values of the weights of a channel tend to be identical, with a low probability of forming a distinctive structure for feature extraction. Similar trends are also observed with the same analysis on MNIST (Fig. A2).

Clearly, setting $\alpha$ to exceed 0.5, the task would not succeed as the first block would be unable to explore the input's rich features effectively. Hence, simply decreasing $ED_c$ cannot facilitate learning. Therefore, we investigate the impact of $ED_c$ by assigning $\alpha = 0.0$ and performing the same experiments as outlined in the previous section to illustrate how task performance is influenced. As shown in Fig. 3**c**, although depending solely on $ED_d$ still results in some learning, the performance is inferior to the standard configuration, particularly for CIFAR10 and CIFAR100. Consequently, optimizing $ED_c$ is an indispensable component for learning.

## 4.3 Ablation study

In standard deep learning, the inference stage typically deactivates dropout and involves a standard feedforward pass through the network, with the network's output serving as the prediction score. In Bayesian neural networks, the inference phase generally utilizes the mean of the outputs for the prediction score. Here, due to the stochastic nature of both the goodness function and the training procedure, we advocate using the average energy of the outputs ($\mathbb{E}[Y^2]$) as the prediction score. We then compare these three inference strategies by running the second training phase with different prediction scores while the trained convolutional blocks are fixed. As shown in Fig. 4**a**, the performance of the proposed method is slightly better than that of the other two methods, indicating that both the mean and the variance can carry information about the labels, while the mean plays the primary role. Interestingly, when we apply t-distributed stochastic neighbor embedding (t-SNE) [34] to visualize the model output as defined by $\mathbb{E}[Y^2]$ (see Fig. A3), the outputs within the same class exhibit a unique fluctuating direction. Moreover, similar classes are not only in close proximity to each other, but also exhibit shared fluctuating directions.

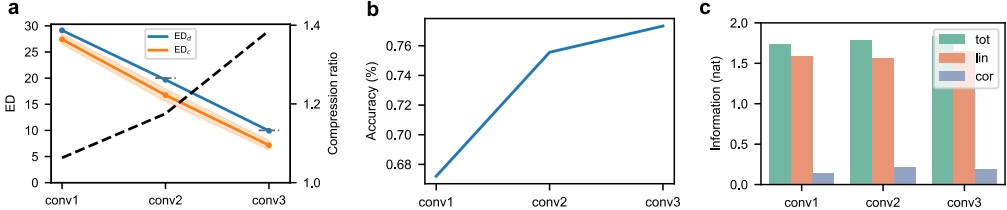

Figure 5: **Layerwise analysis of representations after training a.** Effective dimensionality (ED) of block outputs projected into a lower-dimensional space. $ED_d$ and $ED_c$ are colored by blue and orange respectively and the shades denote one standard deviation of $ED_c$ across classes. The horizontal dashed line marks the projection dimensionality, and the black dashed line shows the compression ratio $ED_d/ED_c$ **b.** Linear separability of each block's representation, measured by training a linear classifier on the output of each block. **c.** Information decomposition of classifier outputs from each block, assuming a Gaussian mixture model. We report total mutual information (*tot*), linearly decodable information*lin*, and second-order interaction terms (*cor*), where *tot = lin + cor*. Results shown are from a randomly selected model trained on CIFAR-10.

We next investigate whether incorporating label information and grouping same-class samples during goodness optimization improve performance. Since our proposed unsupervised method relies on the generation of noisy variants of each sample, we compare it with two supervised alternatives: *Sup+sampling* uses labels and applies the same noise sampling as in the unsupervised setting while *Sup* only uses label information to compute $L$ directly (within-class samples for $ED_c$ and batch data for $ED_d$). As shown in Fig. 4**b**, the unsupervised method slightly outperforms the supervised one, suggesting that the model can learn effectively without labels and achieve comparable accuracy. Notably, noise sampling can improve performance, as *Sup + sampling* is better than *Sup*. Although such a sampling operation increases computational cost, the advance of neuromorphic hardware has the potential to mitigate this by leveraging intrinsic physical noise.

Finally, we investigate the role of the projection scheme by considering three strategies, *Graded* (gradual reduction, as used in the main experiments), *Fixed* (set to the number of classes), and *Random* (arbitrary dimensionality no less than the number of classes), performance remains largely comparable. As shown in Fig. 4**c**, all three projection strategies yield similar performance. However, omitting projection entirely (*None*) leads to a notable performance drop, indicating that projection is essential for task-related performance.

### 4.4 Higher compression ratio leads to better performance

To better understand how the ED is related to task performance, we calculate the $ED_d$ and $ED_c$ of the outputs of each block after projecting it into the same subspace used in training (Fig. 5**a**). Ignoring that each block has a distinct projection dimensionality indicated by the short horizontal dashed lines, we observe that both $ED_d$ and $ED_c$ decrease across the blocks. However, the reduction in $ED_c$ is more pronounced compared to $ED_d$, resulting in an increase in the compression ratio (depicted by the black dashed line). From the perspective of such a lower-dimensional manifold, the input samples belonging to the same class can be gradually explained by fewer dimensions, while samples from different classes are more likely to be explained by different dimensions.

The compression ratio may serve as a valuable indicator of task-related performance efficiency. To demonstrate, we train two linear classifiers on the outputs of the first two blocks respectively, each following the same protocol as that employed for the final block, to assess changes in linear separability across the blocks. As depicted in Fig. 5**b**), performance improves steadily with the addition of more stacked blocks, corresponding to the increase in compression ratio.

However, the gradually increased compression ratio in the projected space does not necessarily imply a similar trend in the original space. In Appendix A1.4, we empirically evaluated the activation sparsity in trained models using Hoyer's sparseness [35] and found that the sparsity may not increase monotonically with depth. In addition, the dataset, projection strategy, and network depth can all affect the activation sparsity.

To better understand how information is represented across layers, we apply information breakdown analysis [36–38] to the output of each block after it passes through the corresponding linear classifier (see Appendix A1.5 for details). We model the classifier output as samples from a Gaussian mixture and estimate the mutual information between these outputs and the class labels. Although the Gaussian assumption may not fully capture the true distribution, it provides insights into the representational structure through moment matching. As shown in Fig. 5c), the mutual information $I_{tot}$ is mainly contributed by the linearly decodable component $I_{lin}$, with the second order interaction $I_{cor}$ contributing modestly. This suggests that while most label-relevant information is accessible to the linear classifier, a small portion remains embedded in neuron-to-neuron correlations and is not linearly separable. We perform a similar analysis on MNIST (Appendix Fig. A4). Given the simplicity of the task, linear separability is nearly saturated across all layers, even as the final block exhibits a higher compression ratio than the previous one. Here, $I_{cor}$ is negligible compared to $I_{lin}$, indicating that almost all the information relevant to the task is captured by linear projections, consistent with the high classification accuracy observed.

## 5 Conclusion and discussion

We have shown that the proposed dimensionality compression loss $L$ enables effective unsupervised learning within the FF framework without requiring negative samples. By injecting noise to create stochastic variants of each input, the model is trained to minimize the effective dimensionality of responses within a class ($ED_c$) while maximizing that across classes ($ED_d$). Experiments on MNIST, CIFAR-10, and CIFAR-100 demonstrate competitive performance using a shallow three-layer CNN, and our ablation studies further indicate that noise benefits learning and that stronger dimensionality compression correlates with better classification accuracy. Despite promising results, our method does not yet achieve state-of-the-art accuracy and has difficulty when scaling to large-scale datasets or deeper architectures. These remain important directions for future research along with the development of biologically plausible implementations on neuromorphic hardware.

The ED objective may also be applied end-to-end and conceptually relates to self-supervised learning methods such as Barlow Twins [39] and VICReg [40]. However, unlike these approaches, which explicitly enforce feature decorrelation or variance–covariance constraints, our method encourages noisy copies of the same sample to reside on a low-dimensional manifold while ensuring distinct representational directions across different inputs.

The high-level principle of our method also relates to predictive coding [41–43], which posits that the brain continuously generates predictions about the incoming sensory input and updates its internal model based on prediction errors. In our framework, minimizing $ED_c$ encourages stable representations that suppress noise-induced variability, whereas maximizing $ED_d$ maintains discriminative information across stimuli. Such a dual objective balances accurate prediction with representational diversity, aligning with recent advances in predictive coding related learning algorithms [24, 44, 45].

There are two main advantages of the proposed method that makes it biological plausible. Firstly, noise is ubiquitous in the brain [7] and can significantly affect the way a neural system represents and processes information [46, 8, 13], an essential characteristic that distinguishes it from deterministic digital computing. In such a stochastic system, we have to measure its outputs multiple times to obtain a reliable estimate. Intuitively, we would like to average the output, which implicitly assumes that variability is just noise that should be ignored. While such firing rate coding is widely adopted in neuroscience, accumulating evidence suggests that variability itself can also be information carriers [11, 47, 48]. Our proposed method uses noise to generate multiple variants of each input sample, which offers an alternative perspective on how the brain may utilize noise and how neuromorphic computing can exploit the intrinsic noise of physical devices to facilitate learning and computation [17, 18, 15, 16, 18].

Secondly, optimizing ED is relatively easy to implement based on biologically feasible mechanisms. For instance, WTA competition is widely observed in various brain regions [49–51], which can be used to reduce $ED_c$ by encouraging a sparse response. Maximizing $ED_d$ is more subtle but still achievable in biologically plausible neural circuits. For example, Bergoin et al. [52] demonstrate that inhibitory neurons and their plasticity can consolidate and selectively separate learned assemblies and limit memory capacity. Future work could explore the design of local learning rules that balance the power of WTA and inhibitory competition to adaptively meet the proposed objective.

## Acknowledgments and Disclosure of Funding

Supported by the National Key R&D Program of China (2019YFA0709502); Supported by National Natural Science Foundation of China (No. 62306078) and the Science & Technology Commission of Shanghai Municipality (No. 25LN3200700); Supported the Ji Hua Laboratory S&T Program (No. X250881UG250) and the Lingang Laboratory (No. LGL-1987); Supported by ZJ Lab and Shanghai Center for Brain Science and Brain-Inspired Technology; Supported by the 111 Project (No. B18015).

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

# Appendix

## A1  Details of the methods

### A1.1  Network architecture

The network architecture used in this work is summarized in Table A1. For all experiments, the setting of the convolutional blocks is the same, and the only difference is the input dimension of the fully connected layer.

Table A1: The details of the network architecture.

| Block index | Components |
|---|---|
| 1 | BatchNorm (No affine) 
 Dropout, $p = 0.2$, creating $N = 20$ noisy copies 
 $5 \times 5$ standard conv, 96 channels, stride 1, padding 2 
 ReLU 
 $4 \times 4$ MaxPooling, stride 2, padding 1 |
| 2 | BatchNorm (No affine) 
 $3 \times 3$ depthwise conv, 384 channels, stride 1, padding 1 
 ReLU 
 $4 \times 4$ MaxPooling, stride 2, padding 1 |
| 3 | BatchNorm (No affine) 
 $3 \times 3$ depthwise conv, 1536 channels, stride 1, padding 1 
 ReLU 
 $2 \times 2$ AvgPooling, stride 2, padding 0 |
| 4 | Flatten 
 Dropout, $p = 0.5$, inplace, for Regularization 
 Linear, 13824 input dims for MNIST, 24576 for CIFAR10 and CIFAR100 |

### A1.2  Details of the experimental design

All models were trained using the AdamW optimizer with a learning rate of 0.001 and a weight decay of 0.01. A cosine annealing learning rate schedule was applied, with a maximum of 3 and 60 iterations for phase 1 and phase 2 training, respectively. The batch size was fixed at 128 across all experiments.

During phase 1 training, given an input batch $X' \in \mathbb{R}^{B \times C \times H \times W}$, after passing through the batch normalization layer of the first convolutional block, we applied dropout with a probability of 0.2 to generate $N = 20$ noisy variants per sample, resulting in ($X \in \mathbb{R}^{B \times N \times C \times H \times W}$). This dropout was only used in the first block and remained active during inference.

The output of each block was projected onto a predefined lower-dimensional space using randomly generated orthogonal basis vectors sampled from the Haar distribution (via SciPy). Under the default *Graded* setting, the projection dimensions were 30-20-10 for MNIST and CIFAR-10 and 90-150-100 for CIFAR-100. The block was then optimized using the proposed $L$ objective computed on the projected output. By default, a trade-off factor $\alpha = 0.5$ was used. To ensure purely local optimization, we detached the output tensors from the computation graph before passing them to the next block, preventing gradient flow across layers.

During the second phase of training, we trained a linear classifier using cross-entropy loss. Due to the sampling in the first block, the classifier output $Y$ had the shape of $B \times 20 \times 10$ for MNIST and CIFAR-10, and $b \times 20 \times 100$ for CIFAR-100. The uncentered second moment across the sampling dimension was used as the prediction score, and the cross-entropy loss was computed against the ground truth labels.

To assess the role of $\alpha$ (Fig 3 **a**-**b** and Fig. A2), we varied $\alpha$ from 0 to 1 in increments of 0.1 when training the first block in MNIST and CIFAR-10 to investigate changes in channel weights. We also

trained models five times with $\alpha = 1.0$ and compared their task performance with the default setting (Fig. 3 **c**).

To study how inference strategies influence performance (Fig 4 **a**), we reused the trained convolutional blocks and retrained the classifier using two alternative strategies. The first used the mean of the output as the prediction score, denoted by $\mathbb{E}[Y]$ while the second used a regular feedforward without the sampling operation.

We also evaluated other projection approaches. In the *Fixed* setting, the projection dimension for each block was matched to the class count. The blocks used a dimension of 10 for MNIST and CIFAR-10. For CIFAR-100, due to channel constraints, the dimensions were 90 for the first block and 100 for the second and third. In the *Random* setup, the projection dimensionality for each block was chosen randomly. For MNIST and CIFAR-10, it varied between 10 and 60; for CIFAR-100, it spanned 100 to 300 (except for the first block, set at 90). In the *None* configuration, the projection operation was omitted.

To minimize the effort of reimplementing Greedy InfoMax (GIM) [24] for comparison, we resued the scripts provided by the authors and modified the network architecture to align with ours, leaving the hyperparameters used for this method unchanged. The training procedure for phase 1 and phase 2 remained the same as in the main experiments except for two differences. Firstly, for the MNIST dataset, to ensure that there were enough patches for prediction, we scaled the input images to $32 \times 32$ while keeping other settings unchanged. Secondly, instead of layer-wise training, the scripts provided by the authors trained all blocks simultaneously in each iteration. For each dataset, we trained models five times with GIM and reported the mean and standard deviation of the test accuracy.

### A1.3 The effect of noise strength and sampling size

To better understand the effect of noise, we conducted experiments by varying the probability of dropout $p$ from 0.1 to 0.5 and the sampling size from 4 to 20 to empirically investigate how the noise level affects performance. Other training settings remained unchanged as in the main experiments. The results are summarized below.

Table A2: The effect of noise strength and sampling size when training on MNIST.

| sample size / $p$ | 0.1 | 0.2 | 0.3 | 0.4 | 0.5 |
|---|---|---|---|---|---|
| 4 | 99.4 | 99.37 | 99.23 | 98.61 | 97.95 |
| 8 | 99.35 | 99.41 | 99.38 | 99.29 | 98.29 |
| 12 | 99.36 | 99.32 | 99.42 | 98.86 | 98.51 |
| 16 | 99.35 | 99.36 | 99.41 | 98.75 | 98.54 |
| 20 | 99.44 | 99.41 | 99.34 | 98.65 | 98.56 |

Table A3: The effect of noise strength and sampling size when training on CIFAR10.

| sample size / $p$ | 0.1 | 0.2 | 0.3 | 0.4 | 0.5 |
|---|---|---|---|---|---|
| 4 | 74.87 | 74.64 | 75.6 | 74.68 | 74.54 |
| 8 | 76.57 | 77.02 | 75.8 | 77.1 | 75.73 |
| 12 | 76.8 | 77.32 | 77.24 | 76.33 | 76.2 |
| 16 | 76.6 | 76.87 | 77.62 | 77.66 | 76.7 |
| 20 | 77.19 | 76.97 | 76.75 | 77.12 | 77.12 |

Table A4: The effect of noise strength and sampling size when training on CIFAR100.

| sample size / $p$ | 0.1 | 0.2 | 0.3 | 0.4 | 0.5 |
|---|---|---|---|---|---|
| 4 | 52.08 | 51.99 | 51.19 | 51.19 | 48.59 |
| 8 | 50.84 | 52.79 | 52.51 | 52.29 | 50.63 |
| 12 | 51.48 | 51.51 | 51.88 | 50.95 | 33.54 |
| 16 | 53.21 | 52.97 | 52.45 | 51.52 | 49.44 |
| 20 | 52.6 | 52.66 | 53.79 | 49.95 | 25.21 |

## A1.4 The sparseness of neural activation

We empirically evaluated the activation sparsity in trained models with Hoyer's sparseness measure:

$$S(x) = \frac{\sqrt{d} - \frac{\|x\|_1}{\|x\|_2}}{\sqrt{d} - 1}, \tag{6}$$

where $x$ is the activation vector of a layer with $d$ neurons. Here, the vector $x$ is the flattened feature map of a convolutional layer from $N$ noisy variants of an input sample and $d$ is the length of $x$. We then calculated the mean and standard deviation of Hoyer's sparseness across five independently trained models for each dataset, with 100 samples per class. The results are summarized in Table A5. Here,

Table A5: Sparseness of neural activation across different layers.

| Layer index | MNIST (FOP/NP) | CIFAR10 (FOP/NP) | CIFAR100 (FOP/NP) |
|---|---|---|---|
| Layer 1 | $0.52 \pm 0.05 / 0.29 \pm 0.06$ | $0.34 \pm 0.06 / 0.47 \pm 0.04$ | $0.55 \pm 0.08 / 0.49 \pm 0.05$ |
| Layer 2 | $0.24 \pm 0.02 / 0.45 \pm 0.07$ | $0.29 \pm 0.05 / 0.38 \pm 0.10$ | $0.42 \pm 0.06 / 0.38 \pm 0.09$ |
| Layer 3 | $0.46 \pm 0.03 / 0.87 \pm 0.04$ | $0.43 \pm 0.04 / 0.54 \pm 0.06$ | $0.39 \pm 0.07 / 0.54 \pm 0.06$ |

FOP means that the model was trained by first projecting the layer's output to a fixed dimensionality using the *Graded* projecting scheme, and NP means directly optimizing the layer's parameters without the projection operation. These results showed that the sparsity varied with datasets, projection strategy, and network depth. It could range from highly sparse (0.87) to quite dense (0.24).

## A1.5 Information breakdown analysis

The mutual information between the target $t$ and the readout $Y$ is defined in terms of the difference between the readout entropy on all stimuli and the conditional entropy for a given class of stimuli as follows.

$$I_{\text{tot}}(Y; t) = h(Y) - h(Y|t). \tag{7}$$

The entropy $h(Y)$ and the conditional entropy $h(Y|t)$ are given by

$$h(Y) = - \int p(Y) \log p(Y) dY, \tag{8}$$

$$h(Y|t) = - \sum_t p(t) \int p(Y|t) \log p(Y|t) dY, \tag{9}$$

where the readout distribution $p(Y|t)$ is modeled as a Gaussian distribution with mean $\hat{\mu}$ and covariance $\hat{\Sigma}$ conditioned on the stimulus class $t$. The readout distribution over all stimuli is calculated as

$$p(Y) = \sum_t p(t)p(Y|t). \tag{10}$$

We used the information breakdown analysis [36] which further dissects mutual information $I_{\text{tot}}$ into three components, allowing us to assess the amount of contributions of individual readout components $I_{\text{lin}}$, signal similarity among readout components $I_{\text{sigsim}}$, and noise correlation in readouts $I_{\text{cor}}$. The quantity $I_{\text{lin}}$ measures the total amount of information that would be transmitted if all readout components were independent, which is given by

$$I_{\text{lin}} = \sum_j [h(Y_j) - h(Y_j|t)], \tag{11}$$

where $Y_j$ is the $j$-th component of the readout. The quantity $I_{\text{sigsim}}$ measures the information loss arising from the redundancy due to overlaps between the tuning curves of each readout component, which is given by

$$I_{\text{sigsim}} = h(Y_{\text{ind}}) - \sum_j h(Y_j), \tag{12}$$

where the independent population response $Y_{\text{ind}}$ is defined by the distribution

$$p(Y_{\text{ind}}|t) = \prod_j p(Y_j|t). \tag{13}$$

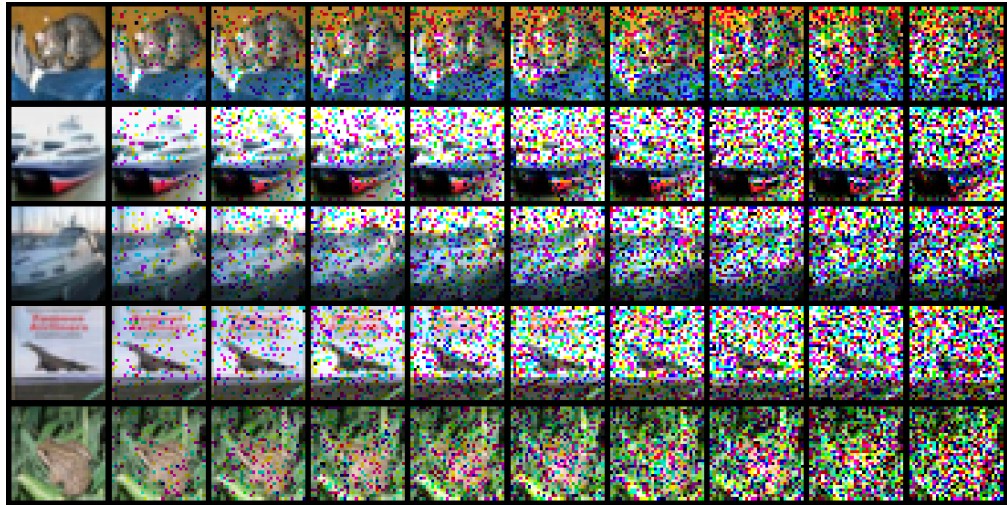

Figure A1: Example of a noisy image from the CIFAR-10 dataset. Each column represents five example images corrupted by randomly dropping its pixels with a probability increment of 0.05.

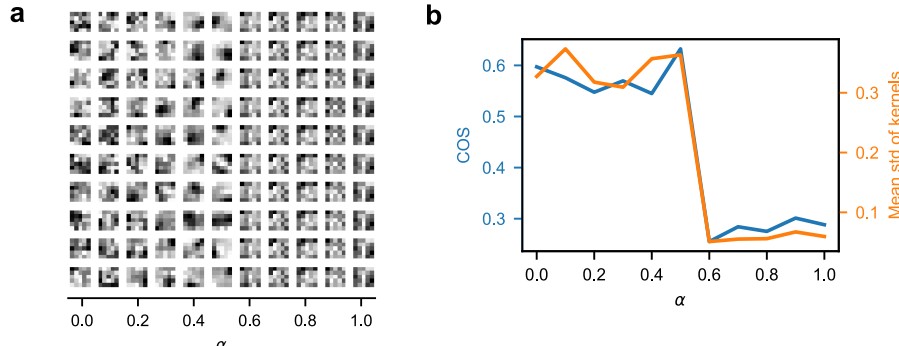

Figure A2: Extension of Fig 3 analysis to MNIST dataset.

The last component $I_{\mathrm{cor}}$ accounts for the rest part of $I_{\mathrm{tot}}$, that is, the total amount of information due to noise correlations in the readout.

$$I_{\mathrm{cor}} = I_{\mathrm{tot}} - I_{\mathrm{lin}} - I_{\mathrm{sigsim}}. \tag{14}$$

For simplicity, we absorb $I_{\mathrm{sigsim}}$ in $I_{\mathrm{lin}}$ and use $I_{\mathrm{lin}}$ to represent the information that can be obtained using linear methods.

We initially used the trained linear classifier shown in Fig. 5b and Fig A4 b to derive the prediction scores $\mathbb{E}[Y^2]$ for each block. Subsequently, these scores were aggregated according to the labels of the inputs, allowing the calculation of their mean and covariance. Using a Gaussian mixture model with 10 components, each sharing the derived mean and covariance, we generated 100,000 samples from each Gaussian component to conduct the information breakdown analysis.

## A2 Visualization and additional analysis

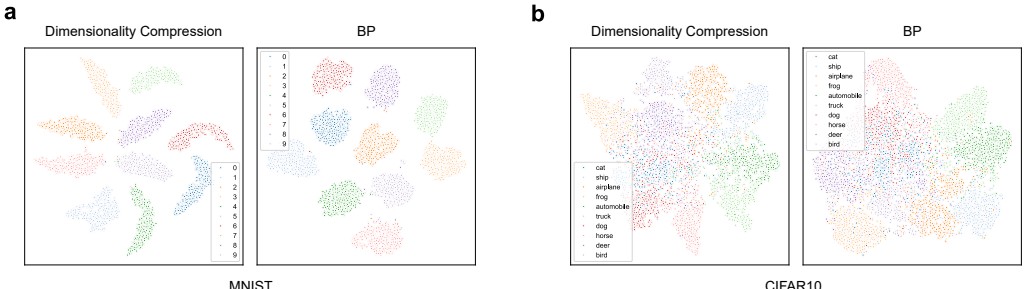

Figure A3: Comparison of outputs of models trained by the proposed method and backpropagation using T-SNE visualization. **a** and **b** are results on MNIST and CIFAR-10 datasets respectively.

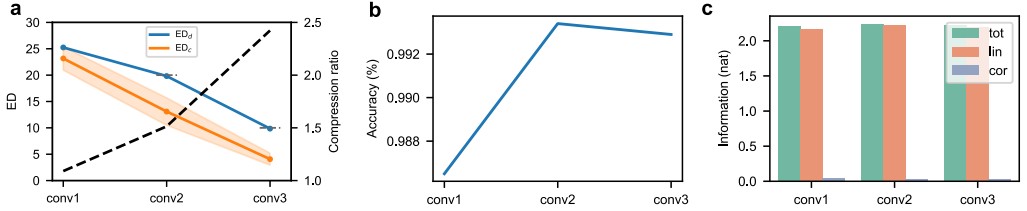

Figure A4: Extension of Fig 5 analysis to MNIST dataset.

