# OpenReview forum: "Stochastic Forward-Forward Learning through Representational Dimensionality Compression"
_NeurIPS.cc/2025/Conference — NeurIPS 2025 poster_

### Official Review · Reviewer_t7od · 2025-06-09

**Clarity:** 3
**Significance:** 3
**Originality:** 3
**Rating:** 4
**Confidence:** 3

**Summary:**

The authors develop a technique to enhance non backpropagation based learning, through the use of contrastive samples that are generated without supervision. Essentially through noise, regularization losses and dimensionality reduction.

**Questions:**

n/a

**Ethical Concerns:**

["NO or VERY MINOR ethics concerns only"]

**Final Justification:**

I hereby confirm my previous review, in that I find it to be an interesting contribution to the field of non gradient learning and biologically plausible alternatives to BP. I hope the authors have the time to run experiments on more complex datasets for the final submission and try to justify with metrics of speed or memory improvements what we gain if we lose a bit of accuracy.

**Limitations:**

Discussed in the conclusions.

**Paper Formatting Concerns:**

Correct.

**Quality:**

3

**Strengths And Weaknesses:**

**Strengths:**
The paper presents a well-motivated and biologically plausible alternative to backpropagation by introducing a novel goodness function based on effective dimensionality (ED). This function captures second-order statistics of neural responses, promoting compact, class-consistent representations and diverse, globally discriminative features. They achieve this without requiring labels or manually crafted negative samples. The use of noise as a constructive mechanism to generate contrastive samples aligns well with principles from neuroscience and neuromorphic computing, providing a practical and theoretically grounded approach for unsupervised learning. Empirical results across MNIST, CIFAR-10, and CIFAR-100 show that the method achieves competitive performance compared to other non-backpropagation algorithms. Furthermore, the inclusion of ablation studies and orthogonality analyses helps support the framework's core claim, that optimizing representational structure via ED leads to more robust and linearly separable features.

**Weaknesses:**
Despite its conceptual elegance, the paper's empirical validation remains relatively limited. The model underperforms compared to equilibrium-based learning and backpropagation, particularly on more complex datasets, and it is tested only on shallow architectures. The reliance on manually chosen hyperparameters (e.g., the projection dimensionality and trade-off weight alpha) and the lack of large-scale or real-world benchmarks weaken the claim of generality. Additionally, while the theoretical motivation is sound, the exposition of experimental results lacks a clear narrative, and some of the key concepts, like the compression ratio and its link to performance, are underexplored in the broader context of learning dynamics. A more direct comparison to EBL and a clearer explanation of what is gained relative to prior methods would further strengthen the contribution.

---

> ### Author Rebuttal · Authors · 2025-07-31
>
> Thank you for your valuable commets. A point-by-point response to the weaknesses and questions are as follows:
>
> >**Weakness1**: Despite its conceptual elegance, the paper's empirical validation remains relatively limited. The model underperforms compared to equilibrium-based learning and backpropagation, particularly on more complex datasets, and it is tested only on shallow architectures.
> >
>
> **Response**:  While we agree that scalability to arbitrary architectures is important, our current experiments show that simply stacking more blocks does not improve performance. In fact, using deeper networks or the same architecture as in EBL leads to worse performance with our ED-based layerwise training compared to shallower models.
>
> This performance drop is likely not due to depth alone. We suspect that this is due to that samples from the same class may not consistently activate the same subset of neurons. As network depth increases, this issue may be amplified, distorting representations and deviating from task-relevant goals. In contrast, EBL benefits from directly clamping outputs to desired labels, which enforces task alignment. We will revise the manuscript to clarify this performance gap compared to EBL. We will revise the manuscript to clarify this performance gap compared to EBL.
>
> Nevertheless, we believe our method can be scaled for deeper networks with enhancements like residual connections.
>
> ---
>
> > **Weakness 2**: The reliance on manually chosen hyperparameters (e.g., the projection dimensionality and trade-off weight alpha) and the lack of large-scale or real-world benchmarks weaken the claim of generality.
> >
>
> While our experiments show the method works across various projection dimensionalities (Fig 4c), we currently use hand-tuned settings because Farrell [c,f] revealed intrinsic ED patterns in BP-trained networks. The optimal configuration for projection dimensionality and $ \alpha $ remains an open question. We acknowledge the need for more rigorous benchmarks and will clarify the scope of our claims regarding generality.
>
> > **Weakness 3**: Additionally, while the theoretical motivation is sound, the exposition of experimental results lacks a clear narrative, and some of the key concepts, like the compression ratio and its link to performance, are underexplored in the broader context of learning dynamics.
> >
>
> **Response**: We acknowledge the lack of theoretical analysis in the original manuscript. Further investigation of the learning dynamics during the weight update process is needed.
>
> > **Weakness4**:  A more direct comparison to EBL and a clearer explanation of what is gained relative to prior methods would further strengthen the contribution.
> >
>
> We will include a more thorough literature review to cover the necessary backgrounds in the revision.
>
> **Reference**
> [c] Farrell, M., Recanatesi, S., Moore, T., Lajoie, G. & Shea-Brown, E. Gradient-based learning drives robust representations in recurrent neural networks by balancing compression and expansion. _Nat Mach Intell_ **4**, 564–573 (2022).
> [f] Farrell, M. Revealing Structure in Trained Neural Networks Through Dimensionality-Based Methods. PhD thesis, Univ. Washington (2020).

---

### Official Review · Reviewer_ck6c · 2025-06-24

**Clarity:** 4
**Significance:** 3
**Originality:** 3
**Rating:** 4
**Confidence:** 4

**Summary:**

The paper describes a local self-supervised learning method. The learning minimizes a loss applied layer-wise, and gradients are blocked across layers. The loss measures the dimensionality of the data using the effective dimension, which is the ratio between the square of the sum of eigenvalues of the covariance matrix, divided by the sum of the squared eigenvalues. The loss is the difference between two forms of ED, one applied to a noisy sample from the same input, and the second is applied across the batch dimension. The dimensionality is therefore maximal for different images in a batch, but minimal with a noisy version of the same image. The method is tested on image classification datasets (with a post-training linear classification accuracy). It performs relatively well in comparison with other layer-wise learning rules. It is reported that the kernel weights become orthogonal through the learning process

**Questions:**

I wonder what the specificity of the chosen loss function is in comparison with other learning algorithms. The algorithm is prominently compared with EBL, but EBL provides an alternative to gradient descent as a whole and is a supervised algorithm. What is the high-level intention of doing this comparison ? References like [1,2,3] are in the sense more similar to the proposed algorithm. If the key question is to provide a bio-plausible learning algorithm, how would he gradient descent step of the ED look in a simple case, would this indeed be apparent to a bio-plausible learning mechanism?

**Ethical Concerns:**

["NO or VERY MINOR ethics concerns only"]

**Final Justification:**

The paper provides a useful analytical approach to layer-wise local learning. The additions promised by the author during the rebuttal will strengthen the paper.

**Limitations:**

Yes

**Paper Formatting Concerns:**

No formatting concerns.

**Quality:**

3

**Strengths And Weaknesses:**

While there have been many different approaches to layer-wise learning rules, I have seen ED being used to analyze data but not yet as a loss function. It is elegant and mathematically sound.
The learning algorithm is mathematically sound and well-explained, and it is great that it uses a simple loss function. The performance is tested on reasonable benchmarks, and the reported performance seems logical and reproducible.

In terms of weaknesses:

- It is unfortunate that the loss function is not tested (with identical architecture and with similar analysis) against alternative loss functions. For instance, I wonder how the result compares to the same architecture but compared with the (more common) InfoNCE loss function. Methods like InfoNCE, which are usually applied end-to-end, can also be applied block-wise [1] and layer-wise [2,3], and vice versa, ED could be applied ETE. So it would be useful to know how ED compares to these methods.

- One of the motivations is to make a "non-contrastive" algorithm. But I suspect that it is secretly a contrastive algorithm: the term ED_d applied on the batch dimension performs comparison on the batch dimension. In fact, it seems that the algorithm does not work for alpha=1, where this term is dropped. So I suspect that the contrastive component is needed. I do not think it is crucial for the message of the paper to claim that the method is non-contrastive, though.

[1] https://arxiv.org/abs/1905.11786
[2] https://arxiv.org/abs/2010.08262
[3] https://www.nature.com/articles/s41593-023-01460-y

---

> ### Author Rebuttal · Authors · 2025-07-31
>
> Thank you for your valuable commets. A point-by-point response to the weaknesses and questions are as follows:
>
> > Weaknesses 1: - It is unfortunate that the loss function is not tested (with identical architecture and with similar analysis) against alternative loss functions. For instance, I wonder how the result compares to the same architecture but compared with the (more common) InfoNCE loss function. Methods like InfoNCE, which are usually applied end-to-end, can also be applied block-wise [1] and layer-wise [2,3], and vice versa, ED could be applied ETE. So it would be useful to know how ED compares to these methods.
> >
>
> **Response**:
>
> Our initial focus was on forward-forward learning, so we primarily reviewed works published after 2022, when the original FF paper [a] was introduced. As a result, we unfortunately overlooked the follow-up work [1].
>
> To address this, we conducted the same experiments using the proposed Greedy InfoMax method, following its architecture and training protocol. The only minor change was resizing input images to 64×64, allowing us to use the authors' hyperparameters without further tuning.
>
> The results are shown below:
>
> |  | Greedy InfoMax | Ours |
> | --- | --- | --- |
> | MNIST | 99.31 | 99.3 $ \pm $ 0.07 |
> | CIFAR10 | 77.94 | 76.96 $ \pm $ 0.73 |
> | CIFAR100 | 50.37 | 53.29 $ \pm $ 1.02 |
>
>
> Due to time and computational constraints, we only ran each experiment once. Nevertheless, the results suggest that our method has the potential to achieve performance comparable to Greedy InfoMax.
>
> While we can also train a network end-to-end using ED, this field has been well studied that there are many successful solutions for unsupervised ETE learning. So we limit our discuss in non-BP methods.
>
> ---
>
> > Weakness 2: - One of the motivations is to make a "non-contrastive" algorithm. But I suspect that it is secretly a contrastive algorithm: the term ED_d applied on the batch dimension performs comparison on the batch dimension. In fact, it seems that the algorithm does not work for alpha=1, where this term is dropped. So I suspect that the contrastive component is needed. I do not think it is crucial for the message of the paper to claim that the method is non-contrastive, though.
> >
>
> **Response**:
> You're correct that $ \text{ED}_d $ implicitly encourages separation between samples, which could be viewed as a contrastive mechanism. However, our key contribution is not about being "non-contrastive" but rather proposing an alternative unsupervised feedforward learning approach that avoids explicit negative sample construction by leveraging noise. We will revise the introduction and Section 3 to clarify this and remove any misleading claims about non-contrastiveness.
>
> ---
>
> > **Question**：I wonder what the specificity of the chosen loss function is in comparison with other learning algorithms. The algorithm is prominently compared with EBL, but EBL provides an alternative to gradient descent as a whole and is a supervised algorithm. What is the high-level intention of doing this comparison ? References like [1,2,3] are in the sense more similar to the proposed algorithm. If the key question is to provide a bio-plausible learning algorithm, how would he gradient descent step of the ED look in a simple case, would this indeed be apparent to a bio-plausible learning mechanism?
> >
> **Response**:
> **Why Compare with EBL?**
> We emphasize connections to Energy-Based Learning (EBL) for three reasons:
>
> 1. Forward-Forward (FF) learning has roots in Boltzmann Machines and Hopfield Networks, making EBL a natural comparison.
> 2.  EBL does not have to be supervised. It can optimize an energy function that similar to FF's layer-wise objective if we restrict the "whole network" as a single layer. Moreover, biological plausible rules such as Hebbian and anti-Hebbian learning, can also be framed as energy minimization by stabilizing correlated activity.
> 3. There are many evidience suppprt that both the mean firing rate and the firing variability can affect neural coding and neural computation [b,f]. In our method, it is naturelly to suggest using $ E[X^2] $ as the metric for classication instead of $ E[X] $, which means we should jointly consider the mean and variance. Interestingly, this fomula also is regard as the energy term in EBL.
>
> Although an explicit weight update rule derived from ED would have non-local terms such as calculating the trace of covariance matrix, we would like to provide the following high-level rationale to support its biological plausibility:
>
> 1. **Plausibility of optimizing ED within local circuits**:
>     - Minimizing $ \text{ED}_c $ naturally leads to a sparse activation pattern, which can be implemented through a biologically plausible winner-takes-all (WTA) mechanism. In such a circuit, only a small subset of neurons respond strongly to a given input, resulting in a low effective dimensionality at the population level due to widespread silencing.
>     - Maximizing $ \text{ED}_d $is more subtle but still achievable in plausible circuits. One scenario involves excitatory and inhibitory neurons operating on different timescales. When an input is clamped, the most active excitatory neurons drive activity in associated inhibitory neurons to maintain network balance. Upon switching from input A to B, the slower inhibitory dynamics can transiently suppress previously active excitatory neurons, allowing other neurons to become active via a renewed WTA competition. This mechanism effectively reshapes the population response to new inputs, thereby increasing $ \text{ED}_d $.
> 2. **Experimental relevance**: The effect of tuning curves [d] and higher-order statistics [e] on neural coding and representational capacity has been extensively studied in neuroscience. The optimization of effective dimensionality offers a unifying theoretical principle that could be tested against neural data in future works.
>
> We will incorporate these points into Section 3 and the Discussion to better explain our motivation and why the proposed objective is compatible with biologically plausible learning mechanisms.
>
> **Reference**
> [a] Hinton, G. The Forward-Forward Algorithm: Some Preliminary Investigations. ArXiv (2022).
> [b] Qi, Y. _et al._ Toward stochastic neural computing. ArXiv (2024)
> [d] Kriegeskorte, N. & Wei, X.-X. Neural tuning and representational geometry. _Nat Rev Neurosci_ **22**, 703–718 (2021).
> [e] Panzeri, S., Moroni, M., Safaai, H. & Harvey, C. D. The structures and functions of correlations in neural population codes. _Nat Rev Neurosci_ (2022)

---

> > ### Comment · Reviewer_ck6c · 2025-08-01
> > **Thank you**
> >
> > Thank you for replying to my review. I feel that my borderline acceptance score is justified and fair.
> >
> > Just a remark regarding the response:
> >
> > > Your circuit hypothesis in the response under "Plausibility of optimizing ED within local circuits:" is interesting.
> >
> > I wish there were some simulation or modeling work to demonstrate that such an inhibitory circuit could indeed implement ED, and its gradient update.
> >
> > > The authors wrote in the response "Our initial focus was on forward-forward learning, so we primarily reviewed works published after 2022, when the original FF paper [a] was introduced. As a result, we unfortunately overlooked the follow-up work [1]."
> >
> > This is a great opportunity to get a more accurate historical picture, but the last statement in this authors' response is also inaccurate; [1] was not a follow-up work of FF. The forward-forward paper is very important, but in fact, it was not the first local unsupervised algorithm working in deep networks. The infomax [1] as a block-wise self-supervised method was published a few years before the FF arxiv came out, Hinton cited [1] in the FF paper. The first layer-wise self-supervised method working in deep networks [2] made connections with brain plasticity and was also published before the FF arxiv.
> >
> > Of course, unsupervised local learning rules may even be traced back further, before it was working on deep networks: Foldiak 1991 https://direct.mit.edu/neco/article/3/2/194/5579/Learning-Invariance-from-Transformation-Sequences Oja 1982 https://link.springer.com/article/10.1007/BF00275687

---

> > > ### Author Response · Authors · 2025-08-05
> > > **Thanks for your comments**
> > >
> > > Thank you for your thoughtful follow-up and for sharing these valuable references. We appreciate your clarification regarding the historical context of local learning algorithms.
> > >
> > > We acknowledge our earlier phrasing may have been misleading and regret the oversight. We will revise the manuscript to more accurately reflect the historical connections of these important contributions and to better situate our work within this broader context.
> > >
> > > Regarding your remark on the circuit hypothesis: we agree that demonstrating such a mechanism through simulation would significantly strengthen the claim. We see this as an exciting direction for future work and appreciate your suggestion.

---

### Official Review · Reviewer_5k7D · 2025-06-27

**Clarity:** 1
**Significance:** 2
**Originality:** 2
**Rating:** 4
**Confidence:** 4

**Summary:**

The article proposes a variation of the forward-forward approach. In the original method, the variance of layer activations is used as a goodness-of-fit measure, with unbounded growth controlled by subtracting the objective evaluated on negative samples. To address the challenge of generating such negative samples, the current article introduces an alternative goodness-of-fit function based on effective dimensionality, defined as the ratio of the trace to the Frobenius norm of class-conditioned correlation matrices of hidden layer activations. The article reports empirical results on MNIST, CIFAR-10, and CIFAR-100 datasets.

**Questions:**

* Section 3, Line 99: " a population of neurons where each
100 neuron is selectively responsive to a particular type of stimuli" is this linear separability condition envisioned for neurrons at all layers, or they are mostly for the final layer neurons?

* Figure 1 (c): Is ED in these figures for the unconditional distributions, or class condition distributions? why should ED of class conditioned distributions depend on the center locations of class clusters?

* What does ED_c in (2) measeure and what does ED_d in (3) measure? why should they be a goodness of fit functions? how do they guarantee that the class conditioned  distribution domains have low effective dimension?

* Does the proposed loss lead to local learning rule? if not what is its advantage over the backpropagation algorithm?

**Ethical Concerns:**

["NO or VERY MINOR ethics concerns only"]

**Final Justification:**

I believe the effective dimensionality–based objective proposed in this article is worthy of attention. My only remaining concern, however, relates to the biological network implementation of this objective. I wanted to clarify this point during the discussion—particularly regarding the WTA-based sparse implementation proposed by the authors in their response. The outcome of that discussion suggests that the biological network architecture and learning dynamics based on the ED objective remain an open question for future research.

**Limitations:**

Yes.

**Paper Formatting Concerns:**

Seems to be compatible with formatting instructions. some hyperlinks do not point to the right possitions (e.g. Figure S1).

**Quality:**

2

**Strengths And Weaknesses:**

## Strengths

- Addressing the negative sample issue of the original forward-forward algorithm is valuable.
- The performance on the benchmark datasets provide improvement over the original forward-forward algorithm.

## Weaknesses

- The article lacks clarity in the description, explanation, and motivation of the proposed approach. The mathematical notation—particularly concerning class conditioning—is imprecise and difficult to follow, which obscures the formulation of the proposed objective. The authors should first clearly define the class-conditioned objective and then demonstrate how it is transformed into its final unsupervised form, explicitly presenting the full loss function across all layers. Additionally, the article should provide a clear rationale for why this objective is expected to be effective.


- The reason why forward-forward algorithm introduced is to provide a biologically plausible explanation for biological networks. It is not clear whether the proposed effective dimensionality loss leads to a biologically plausible structure and local learning rule.

- Although the performance is better than the original forward-forward algorithm it is not better than other biologically plausible algorithms.

The article's presentation requires major improvement. Some examples are:

* Section 2: Background work on forward-forward based and other backprop alternatives are not properly explained.

* Page 2, Line 78-79: DFA stands for  Direct Feedback Alignment. Besides the statement " ...  each block to align its output directly with the final target" for DFA is not very accurate.

* Page 3, Figure 1 caption (b): Incorrect sentence "Within-class responses that colored blue and orange resepectively are tightly clustered, leads to a low ED"

* Page 3, Figure 1 caption (b): There is no gray ellipse visible? is it the  dashed circle?

* Page 3, Figure 1 caption (d): Incorrect sentence "The statistic factors that affecting ED"


* Page 3, Section 3:  The tuning curve has not been properly defined before its use. The authors could use conditional distribution based mathematical notation here.

* Page 3: The authors could provide more motivation about the loss in (1). They can describe why it favors lower dimensions through the norm inequality on the eigenvalues.

* Page 3, Section 3: Equation (1) should be first defined based on label's as it measures the effective dimensionality of the based on the conditional (conditioned on the labels) distributions. This needs to be mathematically precise, so that unsupervised version later defined could be better explained in reference to this definition.

* Page 4: the loss functions in (2)-(4) are not clearly described, and why they should work is not clearly explained.

* Page 14, Figure S1 caption: incorrect sentence " randomly drop its pixes".

---

> ### Author Rebuttal · Authors · 2025-07-31
>
> Thank you for your valuable commets. Due to space limitations, we have focused our response on addressing Weaknesses 1-3, select aspects of Weakness 4, and Questions 1-4 in this rebuttal (while retaining reference numbering). We have also corrected all typographical and grammatical errors mentioned in Weakness 4.
>
> > **Weakness 1**.
>
> **Response**: We understand the mathematical notation in the original manuscript was a bit messy and may be ambiguous or inconsistent at times. Below is the definition of the learning objective with cleaner mathematical notations and a theoretical explanation of why it should work.
>
> Denote $X^{(l)}\in \mathbb{R}^{B\times F}$ the neural response in block $l$ to a batch of noisy input samples (B is the batch size and F is the number of features or neurons). The covariance of the neural responses across random realizations of different inputs is $\Sigma^{(l)}=\mathbb{E}[X^{(l)T}X^{(l)}]$ where $\mathbb{E}[\cdot]$ denotes expectation over random realizations of the inputs. Note that in the noise-free setting the covariance is simply  $\Sigma^{(l)}=X^{(l)T}X^{(l)}$ which captures the data variability. The ED is then defined as
> $\text{ED}(X^{(l)}) = \frac{\text{tr}(\mathbb E[X^{(l)T}X^{(l)}])^2}{\lVert\mathbb E[X^{(l)T}X^{(l)}]\rVert_F^2} = \frac{(\sum_{i=1}^{d} \lambda_i)^2}{\sum_{i=1}^{d} \lambda_i^2},$
> where $\lambda_i$ are the eigenvalues of the uncentered covariance matrix $\mathbb E[X^{(l)T}X^{(l)}]$.
>
> Let $x\in\mathbb{R}^{B\times F}$ be a batch of data samples and use dropout to create its noisy copies, $X$, by randomly setting the elements in x to zero with probability p.
> For the $l$-th block, we denote its output (i.e., the corresponding neural responses) as $X^{(l)}$.  The input block is therefore $X^{(0)}=x$. Further denote the neural responses to a particular input sample in the batch as $X_{i}^{(l)}$, that is, the i-th row of  $X^{(l)}$ .
> The goodness function for each block is then partioned into two terms, a consistency term and a diversity term, defined as:
>
> $\text{ED}_{c}=\frac{1}{B}\sum_i^B\text{ED}\left(X^{(l)}_i\right)$,
>
> and
>
> $\text{ED}_{d}=\text{ED}\left(\mathbb{E}[X^{(l)}]\right)$
>
> where $B$ is the batch size, $\mathbb{E}[X^{(l)}]$ denotes averaging over noisy copies of the input batch. The consistency term describes the average ED of neural responses to noisy realizations of individual input samples and captures the noise variability, whereas the diversity term describes the ED of response distribution across different input samples and captures the data variability.
>
> To understand why our method works, we examine an ideal scenario modeled by a Gaussian mixture model (GMM) with two modes, as in Fig. 1b of the main text. The blue and orange dots represent noisy responses for inputs belonging to classes a and b, respectively. The covariances of class a and class b are visualized by the corresponding blue and orange ellipses, while the overall non-centered covariance of the GMM is depicted by the dashed gray ellipse. In this case, $\mathrm{ED}_c \approx 1$ is computed based on the non-centered covariance within each class, whereas $\mathrm{ED}_d \approx 2$ is computed from the total non-centered covariance across all data.
>
> Furthermore, minimizing $\mathrm{ED}_c$ promotes anisotropy in the statistical moments—i.e., it increases the difference in variance across different dimensions (as shown in Fig.1c). For example, given an input from class $a$, the corresponding response becomes more concentrated along one dimension and more dispersed along another. This asymmetry facilitates linear separability between the two classes.
>
> > **Weakness 2:**
>
> **Response**: Due word limitation, please see the response to the Question mentioned by Reviewer ck6c, where we explain the plausibility of optimizing ED within local circuits.
>
> > **Weakness 3**
>
> **Response:**  We appreciate the reviewer's concern regarding the performance gap between our method and other biologically plausible (non-BP) algorithms. While our method improves upon the original Forward-Forward (FF) algorithm, we acknowledge that it does not yet outperform the best existing non-BP methods (such as EBL).
>
> However, we would like to offer two clarifications:
>
> 1. **Performance among non-BP methods remains comparable**: If we compare only among existing biologically plausible algorithms (excluding our method), their performance generally falls within a similar range. This suggests a broader challenge: non-BP methods as a class still struggle to match the performance of standard BP, likely due to the constraints imposed by biological plausibility (e.g., locality, lack of weight transport).
> 2. Finding **what works** is currently more important than finding **what works best**: At this stage, the priority is to identify learning principles that allow biologically plausible models to function reliably across a variety of tasks. Our contribution lies in demonstrating that optimizing effective dimensionality offers a promising direction that enhances performance within the FF framework while remaining compatible with biological constraints. We believe this lays the groundwork for future improvements in efficiency.
>
> We will revise the Discussion to more explicitly acknowledge this limitation and clarify our intended contribution.
>
> > Weakness 4.4
>
> **Response**: Correct. It refers to the dashed circle, though in general it is an ellipse - depending on the components of the gaussian mixture. Its center represents the mean of the gaussian mixture and the ellipse itself represents the **uncentered covariance** of the gaussian mixture. Note that this does not imply that the gaussian mixture itself has a unimodal shape.
>
> > Weakness 4.6
>
> **Response**: We understand the original manuscript lacks a precise definition of the tuning curve. In our case, if we let $X$ be the noisy neural response to an input sample $x$ with label $c$, then we can write the tuning curve (in mean firing rate) as $\mu(c)=\langle\mathbb{E}[X]\rangle_c$. Here, we use $\mathbb{E}[\cdot]$ to denote expectation over random realizations and $\langle\cdot\rangle_c$ to denote averaging across all data samples with label $c$. One may also write out these expectation explicitly in terms of conditional distribution but we find that would be too cumbersome and it does not improve further the clarity.
>
> > Weakness 4.8
>
> **Response**: The ED in response to Weakness 1 is now more clearly defined. If we restrict the sample data to the same class, then the ${\rm ED}$ formula in the revised notation would measure the effective dimensionality of that class.
>
> **Questions**:
>
> > **Question 1** Section 3, Line 99: " a population of neurons where each 100 neuron is selectively responsive to a particular type of stimuli" is this linear separability condition envisioned for neurrons at all layers, or they are mostly for the final layer neurons?
> >
>
> **Response**:  The statement refers to all layers, not just the final one. This passage provides high-level intuition for why it is possible to define a goodness function suitable for Forward-Forward (FF) learning.
>
> The key idea is that since the final classification depends on the linear separability of the output representations, improving linear separability at each layer—through layerwise optimization—can cumulatively lead to good overall performance. While we do not enforce strict linear separability at every layer, encouraging this property throughout the network helps guide learning in the absence of backpropagation across layers.
>
> We will clarify this point in the text to avoid confusion.
>
> > **Question 2**  Figure 1 (c): Is ED in these figures for the unconditional distributions, or class condition distributions? why should ED of class conditioned distributions depend on the center locations of class clusters?
> >
>
> **Response**:  In this panel, what variable the distribution is conditioned on is irrelevant. It is simply a graphical illustration of how ED would change based on parameters of a distribution - in this case the mean, variance, and correlation of a 2d gaussian.
>
> > Question 3: What does ED_c in (2) measeure and what does ED_d in (3) measure? why should they be a goodness of fit functions? how do they guarantee that the class conditioned distribution domains have low effective dimension?
>
> **Response**:  For the first two questions, please refer to our responses to Weakness 1 and Question 2. Regarding the third question, while we cannot guarantee that class-conditioned distributions will achieve low effective dimensionality (since optimization is unsupervised), we hypothesize that samples from the same class share similar structures, activating the same subset of neurons and thus reducing ED.
>
> Our experiments support this: (a) the method achieves reasonable classification performance, and (b) after training, the ratio $ \text{ED}_d / \langle \text{ED}_c\rangle $ increases, where $ \langle \text{ED}_c\rangle $ is the average class-conditioned ED and $ \text{ED}_d $ is the unconditional ED.
>
> > Question 4 Does the proposed loss lead to local learning rule? if not what is its advantage over the backpropagation algorithm?
>
> **Response**: No. The gradients are still calculated using BP, though there is no gradient flow across blocks. However, this approach could help guide biologically plausible learning rules in future work. For more details, please see response to the Question mentioned by Reviewer ck6c, where we explain the plausibility of optimizing ED within local circuits.
>
> **Reference**
> [d] Kriegeskorte, N. & Wei, X.-X. Neural tuning and representational geometry. _Nat Rev Neurosci_ **22**, 703–718 (2021).
> [e] Panzeri, S., Moroni, M., Safaai, H. & Harvey, C. D. The structures and functions of correlations in neural population codes. _Nat Rev Neurosci_ (2022).

---

> > ### Comment · Reviewer_5k7D · 2025-08-02
> > **Thanks for your response**
> >
> > - I would like to thank the authors for their responses. I have carefully reviewed all the reviews and the corresponding replies. The clarifications provided are helpful, and I hope the authors incorporate them into the main text to avoid potential confusion.  I am planning to increase my rating based on these clarifications.
> >
> > - However, my concern regarding the biological plausibility of ED loss–based learning remains. The goal of the FF algorithm and its variants is to provide a plausible mechanism for learning in biological neural networks, as backpropagation is widely considered biologically implausible. Therefore, it is crucial that the manuscript better communicates how the proposed FF variant aligns with this goal. Without such clarification, the method appears similar in spirit to existing non-contrastive self-supervised approaches—such as Barlow Twins, VICReg etc—which also aim to tighten class clusters while expanding the representation space using augmented inputs, but do not make claims about biological plausibility.
> >
> > On this point, I read the authors’ response to Reviewer ck6c regarding the plausibility of optimizing ED loss within local circuits. I remain unconvinced by the claim that “minimizing $ED_c$ naturally leads to sparse activation patterns.” While $ED_c$ encourages dimensionality reduction, the representations of a class can lie in a low-dimensional subspace without being sparse. Conversely, the use of a WTA mechanism enforces sparsity of individual representation vectors, but does not guarantee that in-class representations occupy a low-dimensional space—unless an additional mechanism is introduced to ensure that different instances of the same class activate overlapping components.
> >
> > Overall, a more detailed account of how ED-based learning could be implemented in a biologically plausible way would strengthen the manuscript and better highlight its contributions.

---

> > > ### Author Response · Authors · 2025-08-05
> > > **Thanks for you comments**
> > >
> > > Barlow Twins is a valuable comparison, as both methods are inspired by neuroscience. At a philosophical level, we think many self-supervised methods implicitly or explicitly encourage representations that are linearly separable, as this is a strong indicator of useful feature learning. To promote separability, similar learning principles are often used. The key technical difference is that Barlow Twins relies on end-to-end training through BP, making it less biologically plausible. In contrast, FF is more aligned with biological constraints, as it relies on local signals  with minimal nonlinearity that avoids backpropagation through layers. Although it does not perfectly meet all criterions of biological plausibility, it offers a promising direction toward mechanisms that can bridge artificial and biological learning [g].
> > >
> > >  Regarding the concern on sparse activation, our discussion is based on the tuning curve assumption illustrated in Fig. 1a and its relevance to neural representations [d]. In this context, minimizing $\text{ED}_c$ favors narrow curves, resulting in a one-hot-like, sparse code.
> > >
> > > We acknowledge that  such tuning curve assumption rarely hold in practice. We thus empirically evaluated the activation sparsity in the trained models with Hoyer’s Sparseness measure [h]:
> > >
> > > $S(a) = \frac{\sqrt{D} - \parallel a\parallel_1/\parallel a \parallel_2}{\sqrt{D} - 1}$
> > >
> > > where $a\in \mathbb R^{D}$ denotes the flattened activation vector extracted from $N$ noisy variants, $C$ channels, and spatial dimensions .
> > >
> > > We then calculated the mean and std of Hoyer's sparseness across different dataset, with 100 samples per class.
> > >
> > > |  | MNIST (FP/NP) | | CIFAR10 (FP/NP) | | CIFAR100 (FP/NP) |
> > > | :---: | :---: | --- | :---: | --- | :---: |
> > > | Layer 1 | $ 0.25\pm 0.03 / 0.14\pm0.03 $ | | $ 0.12\pm 0.03 / 0.29\pm0.03 $ | | $ 0.41\pm0.02 / 0.37\pm0.03 $ |
> > > | Layer 2 | $ 0.10\pm0.01 $/$ 0.22\pm 0.04 $ | | $ 0.17\pm 0.05 $/$ 0.23\pm0.09 $ | | $ 0.19\pm0.06 $/$ 0.24\pm 0.07 $ |
> > > | Layer 3 | $ 0.33 \pm 0.03 $/$ 0.83\pm 0.05 $ | | $ 0.30 \pm 0.04 $/$ 0.37\pm0.05 $ | | $ 0.26\pm0.06 $/$ 0.36\pm 0.06 $ |
> > >
> > >
> > >  Here, **FP** means the model is trained by first project the layer's output to a fixed dimensionality and **NP** means directly optimize the layer's parameter without the projecting operation.
> > >
> > > These results show that sparsity varies with dataset, projection strategy, and network depth. It can range from highly sparse (0.83) to quite dense (0.10). Interestingly, sparsity may not monotonically increase with depth.
> > >
> > > Therefore, the sparsity is influenced not only by the objective function but also by hyperparameters such as projection method, kernel size, and depth.
> > >
> > > We will include these discussions and clarifications in the revised manuscript.
> > >
> > > Regarding the WTA mechanism: We agree that WTA alone is insufficient, but enforcing explicit overlap in activations for samples of the same class may not be necessary. If a class is semantically coherent, its instances will naturally converge toward a shared subset of discriminative features through learning. This aligns with the success of self-supervised methods like contrastive learning, which avoid manual overlap constraints by instead leveraging structured dissimilarity that pushing apart representations of distinct samples while pulling together augmented views of the same instance. A well-designed dissimilarity measurement mechanism such as the proposed excitatory-inhibitory interations, combined with other modification such as implicit regularization, may be suffices to organize representations meaningfully without explicit overlap enforcement.
> > >
> > > We thank you for your insightful comments, which have significantly improved our work.
> > >
> > >
> > >
> > > **Reference**
> > >
> > > [d] Kriegeskorte, N. & Wei, X.-X. Neural tuning and representational geometry. _Nat Rev Neurosci_ 22, 703–718 (2021).
> > >
> > > [g] Ororbia, A. G. Brain-Inspired Machine Intelligence: A Survey of Neurobiologically-Plausible Credit Assignment. ArXiv (2023).
> > >
> > > [h] Hoyer, P. O. Non-negative matrix factorization with sparseness constraints. _Journal of machine learning research_**5**, 1457–1469 (2004).

---

> > > > ### Comment · Reviewer_5k7D · 2025-08-06
> > > >
> > > > I agree that non-contrastive SSL algorithms such as Barlow Twins differ from the proposed approach, primarily because they rely on end-to-end training via backpropagation.  This is mainly due to the  fact that they impose the linear separability condition only to the final representation layer. In contrast, the current work enforces linear separability at all hidden layers, which promotes local learning rather than end-to-end training. It would strengthen the manuscript if the authors could provide neuroscientific or machine learning–based justification for expecting linear separability in intermediate representations.
> > > >
> > > > I am somewhat unclear about the sparsity results mentioned in the response. From what I understand from the sparseness measure table, the authors’ earlier claim that optimizing the ED loss leads to sparse representations does not appear to be consistently supported across different layers and datasets. If this is the case, it may also weaken the proposal for the WTA based biologically plausible ED loss implementation?

---

> > > > > ### Author Response · Authors · 2025-08-08
> > > > >
> > > > > To further clarify the conceptual difference between Barlow Twins and our method, let us recall the Barlow Twins loss function:
> > > > >
> > > > > $ \mathcal L = \sum_i (1-C_{ii})^2 + \lambda \sum_i \sum_{j\neq i} C_{ij}^2 $
> > > > >
> > > > > <font style="color:rgb(36, 36, 36);"> where </font>$ C $<font style="color:rgb(36, 36, 36);">is the cross-correlation matrix computed between the outputs of the two identical networks along the batch dimension:</font>
> > > > >
> > > > > $ C_{ij} = \frac{\sum_b z_{b,i}^A z_{b,j}^B}{\sqrt{\sum_b (z_{b,i}^A)^2}\sqrt{\sum_b(z_{b,j}^B)^2}} $
> > > > >
> > > > > Here $ b $ indexes batch samples, and  $ i, j $ index the output vector dimensions.  $ z^A $and $ z^B $ are the features extracted by the same network for the same inputs, but with different augmentations.
> > > > >
> > > > >  When minimized, the loss enforces   $ C_{ii} = 1 $ and $ C_{ij} = 0, i\neq j $.
> > > > >
> > > > >  This means the learned features are orthogonal and invariant to the applied augmentations, thereby capturing underlying structure in the inputs. In contrast, our method trains the network so that for the same inputs, distorted by noise, the outputs lie on a low-dimensional manifold, with different inputs occupying different dimensions. Orthogonal channel weights can emerge as a by-product of this process, but unlike Barlow Twins, our objective does not explicitly require orthogonality.
> > > > > In addtion,  Barlow Twins do not enforce compressing representational dimensionality.
> > > > >
> > > > >
> > > > > In summary, our method is fundamentally different from Barlow Twins in both concept and methodology. It advances Forward-Forward learning by offering an elegant alternative that bypasses the need for generating negative samples. The use of noise and the formulation of the learning objective make it a promising approach for inspiring biologically plausible learning mechanisms.
> > > > >
> > > > >
> > > > >
> > > > > Regarding **linear separability**, in our framework this also arises as a by-product of minimizing $ \text{ED}_c $ while maximizing $ \text{ED}_d $, based on the  simple tuning curve assumptions. We tested this empirically on real tasks:
> > > > >
> > > > > + For MNIST (Fig. S4), linear separability was already high after the first layer, and adding more layers did not improve it.
> > > > > + For CIFAR-10 (Fig. 5), later layers showed higher linear classifier accuracy compared to the first layer.
> > > > >
> > > > > **On sparsity and WTA:**
> > > > > We agree that sparsity is not uniformly observed across datasets and layers. In practice,   $ \text{ED}_c $ cannot be fully minimized due to its trade-off with maximizing $ \text{ED}_c $.  While our objective remains to minimize  $ \text{ED}_c $ and maximize $ \text{ED}_d $,  the outcome is influenced by the dataset, architecture, and hyperparameters. Importantly, sparsity is not our sole target ($ \text{ED}_c $ ), manifold separation and dimensional selectivity are equally important ($ \text{ED}_d $ ).
> > > > >
> > > > > WTA is a biologically plausible mechanism that reliably reduces  $ \text{ED}_c $ and thus provides a good starting point. However, it should not be considered the only possible circuit implementation. Future work will explore combining WTA with other plausible motifs, such as inhibitory competition to jointly regulate $ \text{ED}_c $ and $ \text{ED}_d $ to produce the desired learning outcomes.  For example,  Bergoin et al. (2023) [i] demonstrate that inhibitory neurons and their plasticity can consolidate and selectively separate learned assemblies and limit memory capacity. This supports the hypothesis that inhibitory competition can be used to regulate intra- and inter-assembly distances, although the effects combining with WTA  need further investigating.
> > > > >
> > > > > [i] Bergoin, R., Torcini, A., Deco, G., Quoy, M. & Zamora-López, G. Inhibitory neurons control the consolidation of neural assemblies via adaptation to selective stimuli. _Scientific Reports_**13**, 6949 (2023).

---

> > > > > > ### Comment · Reviewer_5k7D · 2025-08-08
> > > > > > **Thanks**
> > > > > >
> > > > > > I would like to thank authors for their answers and clarifications. In the light of the  rebuttals and discussions, I will increase my score to 4.

---

> > > > > > > ### Author Response · Authors · 2025-08-09
> > > > > > >
> > > > > > > We sincerely thank the reviewer for the constructive feedback, thoughtful questions, and willingness to engage in discussion throughout the review process. The reviewer’s comments have greatly helped us refine and improve the clarity and scope of our work.

---

### Official Review · Reviewer_RdwQ · 2025-07-01

**Clarity:** 3
**Significance:** 3
**Originality:** 3
**Rating:** 5
**Confidence:** 3

**Summary:**

The authors introduce, in the paper "Stochastic Forward-Forward Learning through Representational Dimensionality Compression", a novel variation of the forward-forward learning algorithm that expands upon the formulation of the layer-wise goodness. The new goodness function is proposed as a measurement of effective dimensionality which goes beyond the mere magnitude goodness that is used typically. This choice is well motivated and allows learning without the need for negative samples and instead cleverly apply noise. Their derivations are supported by experimental evidence on MNIST, CIFAR-10, and CIFAR-100.

**Questions:**

* Can the approach remain effective and competitive for deeper architectures? I.e. also in comparison to the networks used in equilibrium propagation?
* Why are the authors not comparing to layer-wise contrastive methods such as successors of [1]?
* There is no indication about the number of orthonormal basis vectors employed for the pre dimensionality reduction. This is a bit misleading and never is mentioned to sufficient capacity, which could be misleading to readers.
* Can the method be applied to biologically more realistic models both on the neuron and architecture level (recurrent)?
* What would be the effect of changes in the employed noise distribution introduced by dropout?

[1] Löwe, S., O'Connor, P., & Veeling, B. (2019). Putting an end to end-to-end: Gradient-isolated learning of representations. Advances in neural information processing systems, 32.

**Ethical Concerns:**

["NO or VERY MINOR ethics concerns only"]

**Final Justification:**

I believe the authors present a good case for a follow up to the FF algorithm with a more powerful goodness function. They provide, after the rebuttal, a more complete analysis and comparisons. My conclusion is that the paper will benefit the community overall.

**Limitations:**

* Primarlily employing shallow architectures: The authors are encouraged to investigate deeper architectures as this is necessary to properly present the method and strengthen the message
* Although the use of noise is a positive point, it becomes at the same time a question whether the method can work in lower noise regimes too.

The authors are also encouraged to automatically check for typos.

**Paper Formatting Concerns:**

No formatting concerns.

**Quality:**

3

**Strengths And Weaknesses:**

Strengths:

* As far as I know the literature, this dimensionality compression goodness function is novel and original in this setting. Hence, an insightful extension that leverages neuron interactions.
* Clever introduction of noise to enable learning and circumvent the need for negative samples.
* The authors provide sufficient insight in the form of analysis and ablations.

Weaknesses:

* The method effectiveness is demonstrated only on shallow architectures, especially considering that the non-BP methods from the literature consider much deeper networks.
* Equilibrium Propagation: The authors note their performance is behind to that of equilibrium propagation, commenting that it might be a result of deeper networks and increased computation. Based on this the comparison to equilibrium propagation is quite indefinitive and should ideally be tested in a control to compare on equal grounds.
* The dependency on noise could also be harmful. Indeed, what if the neuromorphic device that would be considered offers a lower level of noise than what is needed?

---

> ### Author Rebuttal · Authors · 2025-07-31
>
> Thank you for your valuable commets. A point-by-point response to the weaknesses and questions are as follows:
>
> > **Weaknesss 1**: - The method effectiveness is demonstrated only on shallow architectures, especially considering that the non-BP methods from the literature consider much deeper networks.
> **Weakness 2**: Equilibrium Propagation: The authors note their performance is behind to that of equilibrium propagation, commenting that it might be a result of deeper networks and increased computation. Based on this the comparison to equilibrium propagation is quite indefinitive and should ideally be tested in a control to compare on equal grounds
> **Question 1**: _Can the approach remain effective and competitive for deeper architectures?_
> >
>
> **Response**: Weaknesses 1 and 2, along with Question 1, primarily concern the scalability of our method, so we address them together. While we agree that scalability to arbitrary architectures is important, our current experiments show that simply stacking more blocks does not improve performance. In fact, using deeper networks or the same architecture as in EBL leads to worse performance with our ED-based layerwise training compared to shallower models.
>
> This performance drop is likely not due to depth alone. We suspect that this is due to that samples from the same class may not consistently activate the same subset of neurons. As network depth increases, this issue may be amplified, distorting representations and deviating from task-relevant goals. In contrast, EBL benefits from directly clamping outputs to desired labels, which enforces task alignment. We will revise the manuscript to clarify this performance gap compared to EBL.
>
> Nevertheless, we believe our method can be scaled for deeper networks with enhancements like residual connections.
>
> ---
>
> > Question 2: - **Why are the authors not comparing to layer-wise contrastive methods such as successors of [1]?**
> >
>
> **Response**: Our initial focus was on forward-forward learning, so we primarily reviewed works published after 2022, when the original FF paper [a] was introduced. As a result, we unfortunately overlooked the follow-up work [1].
>
> To address this, we conducted the same experiments using the proposed Greedy InfoMax method, following its architecture and training protocol. The only minor change was resizing input images to 64×64, allowing us to use the authors' hyperparameters without further tuning.
>
> The results are shown below:
>
> |  | Greedy InfoMax | Ours |
> | --- | --- | --- |
> | MNIST | 99.31 | 99.3 $ \pm $ 0.07 |
> | CIFAR10 | 77.94 | 76.96 $ \pm $ 0.73 |
> | CIFAR100 | 50.37 | 53.29 $ \pm $ 1.02 |
>
>
> Due to time and computational constraints, we only ran each experiment once. Nevertheless, the results suggest that our method has the potential to achieve performance comparable to Greedy InfoMax.
>
> ---
>
> > Q3: - **There is no indication about the number of orthonormal basis vectors employed for the pre dimensionality reduction. This is a bit misleading and never is mentioned to sufficient capacity, which could be misleading to readers**.
> >
>
> Thank you for pointing this out. The number of orthonormal basis vectors used for projection was 30-20-10 for MNIST and CIFAR-10 and 90-150-100 for CIFAR-100, as mentioned in the Supplementary Material S1.2.
>
> We will explicitly state the number of orthonormal basis vectors used in the experiments in Section 4 of the main text to avoid any confusion. In Fig. 2, we will add a dimensionality reduction module to clarify the preprocessing step before computing the ED loss.
>
> ---
>
> > Q4: - **Can the method be applied to biologically more realistic models both on the neuron and architecture level (recurrent)?**
> >
>
> Yes. To explore the applicability of our method to more biologically realistic models, we conducted experiments using the Moment Neural Network (MNN) [b], which captures the dynamics of firing statistics of spiking neural network (SNN) up to second order moments. Given noisy inputs with first and second moments $ (\mu,\Sigma) $, the MNN produces corresponding spike count moments $ (\hat\mu,\hat C)=\phi(\mu,C $) through a nonlinear mapping.
>
> In this framwork, $ ED_c $ can be computed directly.  For $ ED_d $ , we define the non-centered total covariance over a batch as:
> $ \mathbb E[XX^T] = \frac{1}{B}\sum_{i=1}^B (C_i + \mu_i\mu_i^T) $
> We tested this setup on MNIST using a one-hidden-layer MNN and achieved 92% accuracy. Although this is below state-of-the-art, it demonstrates that a biologically realistic model (SNNs) can potentially be optimized in a similar fashion due to the close correspondence between MNN and SNN.
>
> Regarding recurrent networks, we were unable to conduct experiments due to time constraints. However, prior empirical findings by Farrell et al. [c] show that ED in RNNs increases and then decreases during training with backpropagation. This suggests that optimizing ED in a principled way could yield competitive results in RNNs as well.
>
> > **Weakness 3** ：- The dependency on noise could also be harmful. Indeed, what if the neuromorphic device that would be considered offers a lower level of noise than what is needed? - What would be the effect of changes in the employed noise distribution introduced by dropout?
> **Question 4**: - What would be the effect of changes in the employed noise distribution introduced by dropout?
> >
>
> To better understand the effect of noise, we conducted experiments by varying the dropout probability (from 0.1 to 0.5) and the sampling size (from 4 to 20) to empirically investigate how the noise level affects performance. The results are summarized below:
>
> **MNIST**
>
> | sample size / P | 0.1 | 0.2 | 0.3 | 0.4 | 0.5 |
> | --- | --- | --- | --- | --- | --- |
> | 4 | 99.4 | 99.37 | 99.23 | 98.61 | 97.95 |
> | 8 | 99.35 | 99.41 | 99.38 | 99.29 | 98.29 |
> | 12 | 99.36 | 99.32 | 99.42 | 98.86 | 98.51 |
> | 16 | 99.35 | 99.36 | 99.41 | 98.75 | 98.54 |
> | 20 | 99.44 | 99.41 | 99.34 | 98.65 | 98.56 |
>
>
> **CIFAR10:**
>
> | **sample size / P** | **0.1** | **0.2** | **0.3** | **0.4** | **0.5** |
> | --- | --- | --- | --- | --- | --- |
> | **4** | 74.87 | 74.64 | 75.6 | 74.68 | 74.54 |
> | **8** | 76.57 | 77.02 | 75.8 | 77.1 | 75.73 |
> | **12** | 76.8 | 77.32 | 77.24 | 76.33 | 76.2 |
> | **16** | 76.6 | 76.87 | 77.62 | 77.66 | 76.7 |
> | **20** | 77.19 | 76.97 | 76.75 | 77.12 | 77.12 |
>
>
> **CIFAR100**
>
> | **sample size / P** | **0.1** | **0.2** | **0.3** | **0.4** | **0.5** |
> | --- | --- | --- | --- | --- | --- |
> | **4** | 52.08 | 51.99 | 51.19 | 51.19 | 48.59 |
> | **8** | 50.84 | 52.79 | 52.51 | 52.29 | 50.63 |
> | **12** | 51.48 | 51.51 | 51.88 | 50.95 | 33.54 |
> | **16** | 53.21 | 52.97 | 52.45 | 51.52 | 49.44 |
> | **20** | 52.6 | 52.66 | 53.79 | 49.95 | 25.21 |
>
>
> These results demonstrate that both noise strength (dropout probability) and sampling size significantly influence model performance. In general, moderate noise levels and appropriate sampling sizes yield optimal results.
>
> Beyond the numerical findings, we offer further commentary on this issue. Currently, digital computing hardwares often aim at eliminating noise. However, neuromorphic chips are designed to closely mimic the computational mechanisms of the brain, one of which is the highly irregular and stochastic nature of spiking activity. This inherent variability—often regarded as noise—has been shown to play a functional role in biological computation. Consequently, many studies explore how the brain harnesses such stochasticity to perform useful work. In the long term, we anticipate that neuromorphic hardware will likely evolve toward analog computation, embracing rather than suppressing intrinsic noise. In this context, the noise-dependent optimization paradigm may not be a limitation, but a reflection of biological realism and a design opportunity.
>
> ---
>
> **Reference**
> [a] Hinton, G. The Forward-Forward Algorithm: Some Preliminary Investigations. ArXiv (2022).
> [b] Qi, Y. _et al._ Toward stochastic neural computing. ArXiv (2024).
> [c] Farrell, M., Recanatesi, S., Moore, T., Lajoie, G. & Shea-Brown, E. Gradient-based learning drives robust representations in recurrent neural networks by balancing compression and expansion. _Nat Mach Intell_ **4**, 564–573 (2022).

---

> > ### Comment · Reviewer_RdwQ · 2025-08-06
> >
> > I would like to thank the authors for their detailed rebuttal and conducting additional experiments. Virtually all my concerns have been addressed sufficiently, regarding comparisons with other methods and questions about noise. This, along with careful consideration of the other reviews, justifies an increase in my score, also to appreciate that the community is likely to benefit from this work.
> >
> > To this end, I believe the biggest question mark and opportunity is still whether one can consistently improve with increasing depth.

---

> > > ### Author Response · Authors · 2025-08-08
> > > **Thanks for your comment**
> > >
> > > We fully agree that understanding how to consistently improve performance with increasing network depth is a key open question and an important future direction for FF learning. We plan to systematically explore this in future work.

---

### Decision · Program_Chairs · 2025-09-17

**Decision:**

Accept (poster)

**Comment:**

The paper describes a local self-supervised learning method based on a variation of the forward-forward approach.
The loss used is based on the dimensionality of data, using the effective dimension, which is the ratio between the square of the sum of eigenvalues of the covariance matrix, divided by the sum of the squared eigenvalues.
The dimensionality is maximal for different images in a batch, but minimal with a noisy version of the same image.


Strengths identified by the reviewers include:

1. The dimensionality compression goodness function is novel and original in this setting.

2. Performance on the benchmark datasets provide improvement over the original forward-forward algorithm.

3. The introduction of noise to enable learning and circumvent the need for negative samples is of interest.


Weaknesses identified, which the authors should try to address, include:

1. Algorithm demonstrated only on shallow architectures.

2. The mathematical notation is imprecise.

3. Although the performance is better than the original forward-forward algorithm it is not better than other biologically plausible algorithms.

The authors in their rebuttal did a good job explaining some misunderstandings regarding biological plausibility and the effectiveness of the method with deeper architectures. This convinced some of the reviewers to upgrade their score.

After rebuttal, the paper received 1 Accept, and 3 Borderline accepts. As such I am inclined to recommend acceptance of the paper. The authors should make a reasonable effort to address the concerns of the reviewers in the final paper version.